# Identification of Key Genes during Ethylene-Induced Adventitious Root Development in Cucumber (*Cucumis sativus* L.)

**DOI:** 10.3390/ijms232112981

**Published:** 2022-10-26

**Authors:** Yuzheng Deng, Chunlei Wang, Meiling Zhang, Lijuan Wei, Weibiao Liao

**Affiliations:** 1College of Horticulture, Gansu Agricultural University, Lanzhou 730070, China; 2College of Science, Gansu Agricultural University, Lanzhou 730070, China

**Keywords:** transcriptome, carbon metabolism, secondary metabolism, plant hormone signal transduction

## Abstract

Ethylene (ETH), as a key plant hormone, plays critical roles in various processes of plant growth and development. ETH has been reported to induce adventitious rooting. Moreover, our previous studies have shown that exogenous ETH may induce plant adventitious root development in cucumber (*Cucumis sativus* L.). However, the key genes involved in this process are still unclear. To explore the key genes in ETH-induced adventitious root development, we employed a transcriptome technique and revealed 1415 differentially expressed genes (DEGs), with 687 DEGs up-regulated and 728 DEGs down-regulated. Using Kyoto Encyclopedia of Genes and Genomes (KEGG) pathway analysis, we further identified critical pathways that were involved in ETH-induced adventitious root development, including carbon metabolism (starch and sucrose metabolism, glycolysis/gluconeogenesis, citrate cycle (TCA cycle), oxidative phosphorylation, fatty acid biosynthesis, and fatty acid degradation), secondary metabolism (phenylalanine metabolism and flavonoid biosynthesis) and plant hormone signal transduction. In carbon metabolism, ETH reduced the content of sucrose, glucose, starch, the activity of sucrose synthase (SS), sucrose–phosphate synthase (SPS) and hexokinase (HK), and the expressions of *CsHK2*, *pyruvate kinase*
*2* (*CsPK2*), and *CsCYP86A1*, whereas it enhanced the expressions of *β-amylase 1* (*CsBAM1*) and *β-amylase 3* (*CsBAM3*). In secondary metabolism, the transcript levels of *phenylalanine ammonia-lyase* (*CsPAL*) and *flavonoid 3′-monooxygenase* (*CsF3′M*) were negatively regulated, and that of *primary-amine oxidase* (*CsPAO*) was positively regulated by ETH. Additionally, the indole-3-acetic acid (IAA) content and the expressions of auxin and ETH signaling transduction-related genes (*auxin transporter-like protein 5* (*CsLAX5*), *CsGH3.17*, *CsSUAR50*, and *CsERS*) were suppressed, whereas the abscisic acid (ABA) content and the expressions of ABA and BR signaling transduction-related genes (*CsPYL1*, *CsPYL5*, *CsPYL8*, *BRI1-associated kinase 1* (*CsBAK1*), and *CsXTH3*) were promoted by ETH. Furthermore, the mRNA levels of these genes were confirmed by real-time PCR (RT-qPCR). These results indicate that genes related to carbon metabolism, secondary metabolite biosynthesis, and plant hormone signaling transduction are involved in ETH-induced adventitious root development. This work identified the key pathways and genes in ETH-induced adventitious rooting in cucumber, which may provide new insights into ETH-induced adventitious root development and will be useful for investigating the molecular roles of key genes in this process in further studies.

## 1. Introduction

An adventitious root is a kind of lateral root that often arises from leafstalk, stems, and some non-pericycle tissue [1,2]. The developmental process of an adventitious root is complicated and can be divided into three successive phases: induction, formation, and expression [3]. Adventitious rooting represents the regenerative ability of plants and these roots can obtain water and nutrients from the soil, fix plants on the substrate, and store food [3]. Hence, adventitious rooting has a pivotal role in vegetative propagation. Adventitious root development is influenced by various environmental and endogenous factors, such as mineral nutrition, light, temperature, ectomycorrhizas, aging, phenolic compounds, and phytohormones [4].

Ethylene (ETH), a gaseous signaling phytohormone, is involved in many processes of plant growth and development, including seed dormancy and germination [5], seedling growth [6], primary root development [7], adventitious root formation [8], sex expression [9], fruit set [10], fruit firmness [11], and fruit and vegetable preservation [12]. Moreover, ETH is a key stress signal molecule that responds to many adverse ecological conditions, including cold and freezing stress, heat stress, drought stress, flooding, salt stress, and heavy metal stress [13,14].

Although ETH may influence many aspects of plant growth and development, the roles of ETH in adventitious root development are still unclear. Our previous studies have demonstrated that 10 μM ETH could significantly induce adventitious root development in cucumber and marigold (*Tagetes erecta* L.) [1,15,16]. Moreover, ETH is associated with other signal molecules, such as nitric oxide (NO), auxin, and calcium ions (Ca^2+^) in the promotion of adventitious rooting [1,8,17,18]. For example, NO could be involved in ETH-promoting adventitious root development by enhancing the activities of nitric oxide synthase (NOS) and nitrate reductase (NR), and by regulating rooting-related genes [*CsDNAJ-1* and *calcium-dependent protein kinases 1/5* (*CsCDPK1/5)*] [1]. The application of indole-3-butyric acid (IBA) could promote the adventitious rooting in apple (*Malus domestica* Borkh.) rootstock by regulating ETH synthesis. Both auxin and ETH inhibitors significantly prevented adventitious rooting in apple [17]. Auxin increased the transcript levels of ETH biosynthesis genes, including 1-aminocyclopropane-l-carboxylic acid synthase (ACS) and ACC oxidase (ACO), which resulted in the retention of ETH [8]. Therefore, auxin could trigger the development of an adventitious root through the ETH synthesis pathway. Under salt stress, ETH may be involved in adventitious rooting induced by Ca^2+^ [18]. The application of calcium chloride (CaCl_2_) enhanced the activities of ETH synthesis enzymes and increased the content of ETH precursor 1-aminocyclopropane-1-carboxylic acid (ACC) and endogenous ETH under salt stress, indicating that ETH might act as the downstream molecule in Ca^2+^-induced adventitious root development in cucumber.

ETH and its crosstalk with other signaling molecules could regulate adventitious root development. However, many studies have only elucidated its mechanism from the perspective of pharmacology, and the genes or proteins in ETH-induced adventitious rooting are still unknown. Proteomic analysis found that exogenous ETH could promote adventitious rooting in cucumber by regulating some important proteins [16]. However, the key genes of ETH-induced adventitious rooting are still unclear. Transcriptome, an effective genetic research technique, has been widely applied to identify and analyze some key genes in plants and animals. The method has the characteristics of low cost, high efficiency, and high precision. [19]. Hence, to illustrate the key genes affecting ETH in adventitious root development in cucumber, the transcriptome technique was used in the present study. Using KEGG pathway analysis, some critical differentially expressed genes (DEGs) in enriched pathways, such as carbon metabolism, secondary metabolism, and plant hormone signal transduction, were analyzed. To verify the reliability of the transcriptome data, the contents of sucrose, glucose, starch, and endogenous indole-3-acetic acid (IAA) and abscisic acid (ABA), the activity of enzymes, including sucrose synthase (SS), sucrose–phosphate synthase (SPS), hexokinase (HK), and phenylalanine ammonia-lyase (PAL), and the expressions of key genes were determined. Therefore, this work may provide new insights into ETH-induced adventitious root development and be useful for further study of ETH-prompted rooting at the molecular level.

## 2. Results

### 2.1. DEGs, GO, and KEGG Analysis

In this study, we identified a total of 1415 DEGs in the cucumber transcriptome under ETH treatment, among which 687 DEGs were up-regulated and 728 DEGs were down-regulated (Figure 1A and Appendix A). These DEGs are specific to the ETH-induced adventitious root development process. The number of non-ETH-induced adventitious rooting DEGs was 19,256 (Figure 1A). GO annotation of the DEGs showed that 634 terms were classified as BP, CC, and MF (Figure 1B, Appendix A). In the BP category, the “carbohydrate catabolic process (11 DEGs)”, “nucleotide catabolic process (nine DEGs)”, “nucleoside phosphate catabolic process (nine DEGs)”, “nucleobase-containing small molecule biosynthetic process (10 DEGs)” and “pyruvate metabolic process (nine DEGs)” were enriched. In the CC category, the “extracellular region (seven DEGs)”, “cell wall (10 DEGs)”, and “external encapsulating structure (10 DEGs)” were enriched. In the MF category, “hydrolase activity, hydrolyzing O-glycosyl compounds (30 DEGs)”, “protein heterodimerization activity (11 DEGs)”, “hydrolase activity, acting on glycosyl bonds (30 DEGs)”, “sulfate transmembrane transporter activity (four DEGs)”, and “sulfur compound transmembrane transporter activity (four DEGs)” were the top five enriched terms (Figure 1B).

The top 20 KEGG pathways corresponding to the most abundant DEGs are displayed in Table 1. Among them, “linoleic acid metabolism (6 DEGs)”, “starch and sucrose metabolism (17 DEGs)”, “sesquiterpenoid and triterpenoid biosynthesis (5 DEGs)”, “fatty acid degradation (8 DEGs)”, “isoquinoline alkaloid biosynthesis (4 DEGs)”, “phenylalanine metabolism (7 DEGs)”, “plant hormone signal transduction (26 DEGs)”, “fatty acid metabolism (9 DEGs)”, “pentose and glucuronate interconversions (10 DEGs)” and “cutin, suberin and wax biosynthesis (5 DEGs)” were the top 10 enriched terms in the KEGG pathways (Table 1 and Appendix A).

### 2.2. Carbon Metabolism

#### 2.2.1. Starch and Sucrose Metabolism

As shown in Figure 2A, 17 DEGs and 1 DEP were detected in starch and sucrose metabolism (Appendix A). In the sucrose metabolism pathway, one hexokinase (HK) gene, *HK2* (*LOC101216058*), one beta-glucosidase (β-GC) gene named *beta-glucosidase 11-like* (*LOC101214838*), and three cellulose genes, *endoglucanase 17* (*eg17*; *LOC101204921*), *endoglucanase 6* (*eg6*; *LOC101203271*), and *endoglucanase 9* (*eg9*; *LOC101203271*), were down-regulated by ETH treatment. Moreover, exogenous ETH suppressed the transcript level of two sucrose synthesis-related genes *sucrose synthase* (*SS*; *LOC101211461*) and *sucrose-phosphate synthase* (*SPS*; *LOC101221022*), and two trehalose synthesis-related genes *trehalose phosphatase* (*TP*; *LOC101211097* and *LOC101212172*). Furthermore, the transcript levels of three *glucose-1-phosphate adenylyltransferase* (*G1PA*) genes (*LOC101221562*, *LOC101208904*, and *LOC101208605*) were down-regulated by ETH treatment (Figure 2A).

In starch metabolism, the *1,4-alpha-glucan branching enzyme* (*GBE1*) gene (*LOC101215687*) for starch synthesis in ETH was up-regulated. ETH treatment down-regulated the expression of two *beta-amylase* (*β-Amy*) genes (*LOC101221739* and *LOC101213455*) for dextrin synthesis and up-regulated that of the *β-Amy* gene (*LOC101222005*). The transcript level of *isoamylase* (*ISA*; *LOC 101209348*) was decreased (Figure 2A and Appendix A).

In order to verify the reliability of the transcriptome data, we measured the sucrose, glucose, and starch contents, relevant enzyme activity, and gene expression during ETH-induced adventitious rooting in cucumber. As shown in Figure 2B, compared with the control, ETH treatment significantly decreased the sucrose, glucose, and starch contents in cucumber explants. Moreover, the activities of SS, SPS, and HK with ETH treatment were significantly lower than those in the control (Figure 2C). Additionally, the expressions of six DEGs were determined by quantitative real-time PCR (RT-qPCR) assays (Figure 2D). Among them, except for *CsGBE1*, the expressions of other genes, including *Cseg9*, *Cseg6*, *Csβ-G11-like*, *CsHK2*, and *CsSS5-like* were decreased with ETH treatment.

#### 2.2.2. Glycolysis/Gluconeogenesis, Citrate Cycle (TCA Cycle) and Oxidative Phosphorylation

Twelve DEGs in the glycolysis/gluconeogenesis pathway were identified (Figure 3A and Appendix A). The *6-phosphofructokinase* (*6-PFK*) gene (*LOC101202739*) and *diphosphate-fructose-6-phosphate 1-phosphotransferase* (*DF-6P-1P*) gene (*LOC101215219*) were both down-regulated in ETH treatment. Additionally, two *phosphoglycerate kinase* (*PGK*) genes (*LOC101219631* and *LOC101215513*) were up-regulated and down-regulated, respectively. Two pyruvate biosynthesis-related genes *pyruvate kinase* (*PK*; *LOC101206620* and *LOC101220786*) were down-regulated. The relative expressions of *pyruvate dehydrogenase* (*PDG*; *LOC101204247*) and *dihydrolipoyllysine-residue acetyltransferase* (*DRAT*; *LOC101215151* and *LOC101220730*) were down-regulated by ETH. Meanwhile, ETH increased the transcript level of *aldehyde dehydrogenase* ((*NAD^+^*) *ALDH*; *LOC101219569*), whereas it decreased the transcript level of *alcohol dehydrogenase* (*ADH*; *LOC101211996*).

In the TCA cycle, five DEGs were detected. Among them, one *PDG* and two *DRAT* genes appeared in the glycolysis pathway (Figure 3B and Appendix A). Exogenous ETH down-regulated the expression of *ATP citrate synthase* (*CS*; *LOC101209539*) for citrate synthesis and *isocitrate dehydrogenase NADP^+^* (*IDH*; *LOC101202770*).

There were four DEGs in the oxidative phosphorylation pathway (Figure 3C and Appendix A). Among them, the transcript levels of *NADH dehydrogenase [ubiquinone] 1 alpha subcomplex subunit 2* (*NDUFA2*; *LOC 101208633*) and *NADH dehydrogenase subunit 5* (*nad*5; *LOC11123919*) in complex I were enhanced. In complex V, the expression of uncharacterized gene *LOC101205555* was down-regulated, whereas the expression of *plasma membrane ATPase 4* (*PMA4*; *LOC101213505*) was up-regulated by ETH treatment.

Compared with the control, ETH treatment significantly decreased PK activity (Figure 3D). Moreover, the relative expression of *Csnad5* with ETH treatment was remarkably enhanced in comparison with the control, whereas the relative expressions of *CsPFP*, *CsPK2*, *CsACS2*, and *CsIDH* were decreased (Figure 3E). These results are consistent with those of the transcriptome analysis.

#### 2.2.3. Fatty Acid Biosynthesis and Fatty Acid Degradation

Totals of four and eight DEGs were mapped to the pathways of fatty acid biosynthesis and fatty acid degradation, respectively (Figure 4A and Appendix A). In the fatty acid biosynthesis pathway, ETH increased the transcript levels of *FabG* (*LOC101221009*) and *long-chain-fatty-acid-CoA ligase* (*ACSL*; *LOC101207461* and *LOC101212158*), whereas it down-regulated the expression of *oleoyl-[acyl-carrier-protein] hydrolase* (*OLAH*; *LOC101210955*). In the fatty acid degradation pathway, except for *ADH* and *long-chain fatty acid omega-monooxygenase* (*CYP86A*), the other gene expressions were up-regulated by ETH treatment (Figure 4A).

According to the RT-qPCR analysis, the relative expression of *CsACSL6* with ETH treatment had a significant increase compared with the control, whereas the relative expression of *CsCYP86A1* in the pathway of fatty acid degradation significantly declined with ETH treatment in comparison with the control (Figure 4B).

### 2.3. Secondary Metabolism

#### 2.3.1. Phenylalanine Metabolism

Seven DEGs were detected in the phenylalanine metabolism (Figure 5A and Appendix A). Among them, the transcript levels of two *amidase* genes (*LOC101214819* and *LOC101204792*) were up-regulated and down-regulated, respectively. ETH treatment up-regulated the relative expression of *aspartate transaminase* (*AST*; *LOC101217299*) and three *primary-amine oxidase* (*PAO*) genes (*LOC101210998*, *LOC101213059*, and *LOC101212818*). Moreover, the expression of *phenylalanine ammonia-lyase* (*PAL*; *LOC101218856*) for phenylalanine synthesis was down-regulated by ETH treatment (Figure 5A).

As shown in Figure 5B, the activity of PAL with ETH treatment was significantly lower than that of the control. The relative expression of *CsPAL* was decreased, whereas the expression of *CsPAO* was enhanced by ETH treatment (Figure 5C).

#### 2.3.2. Flavonoid Biosynthesis

In flavonoid biosynthesis, except for shikimate O-hydroxycinnamoyltransferase (HCT; LOC101207100), the transcript levels of chalcone isomerase (CHI; LOC101218716) and flavonoid 3′-monooxygenase (F3′M; LOC101207160) were down-regulated by ETH treatment (Figure 6A and Appendix A). The relative expression of CsF3′M was decreased by ETH treatment, whereas the expression of CsHCT was significantly increased (Figure 6B).

### 2.4. Plant Hormone Signal Transduction

Twenty-six DEGs and one DEP were identified in the plant hormone signal transduction pathway in total (Figure 7A and Appendix A). There were eight DEGs in the auxin signaling pathway in total. Among them, ETH treatment down-regulated the expression of two auxin transporter genes *AUXIN 1 (AUX1; LOC101205750* and *CS-AUX1*) (Figure 7A). The expression of three *auxin/indole-3-acetic acid* (*AUX/IAA*) genes (*LOC101218948*, *LOC101215306*, and *LOC101223028*) was up-regulated and one *AUX/IAA* gene (*LOC10120997*) was down-regulated by ETH treatment. The transcript levels of IAA-amido synthetase *GH3* (*LOC101206247*) and auxin-responsive gene *small auxin up-regulated RNA* (*SAUR*; *LOC101217643*) were decreased by ETH treatment. In the cytokinin (CTK) signaling pathway, *CTK response 1* (*CRE1*; *LOC101209777*) was down-regulated by ETH. Moreover, two DEGs were involved in ETH-induced adventitious rooting in the gibberellin (GA) signaling pathway. The expression of the *DELLA* gene (*LOC101213450*) in the gibberellin (GA) signaling pathway was down-regulated by ETH, whereas the *TF* gene (*LOC101205993*) was up-regulated. In the pathway of abscisic acid (ABA) signaling, six DEGs were found. The expressions of three ABA receptors *PYR/PYL* (*LOC101208823*, *LOC101204882*, and *LOC101210262*), two *SNF1-related protein kinase 2* (*SnRK2*; *LOC101209540* and *LOC101214088*), and the *ABA-responsive element* (*ABRE*)*-binding factor* (*ABF*) gene (*LOC101222720*) were up-regulated. Furthermore, four DEGs were displayed in the ETH signaling transduction pathway. The transcript levels of ETH receptor *ETR* (*CS-ERS*), *EIN3-binding F-box1 and 2* (*EBF1/2*; *LOC101207154*), and two *ERF1/2* (*LOC101206564* and *LOC101220325*) genes were decreased by ETH (Figure 7A). In the brassinosteroid (BR) signaling transduction pathway, two DEGs were involved in ETH-triggered AR development. The expressions of the *BRI1-associated kinase 1* (*BAK1; LOC101216386*) and *touch 4* (*TCH4*; *XTH3*) genes were up-regulated by ETH. The jasmonic acid (JA) signaling pathway-related gene *jasmonate resistant 1* (*JAR1*; *LOC101212037*) was up-regulated by ETH treatment. Additionally, ETH up-regulated the expression of two *TGACG-binding* (*TGA*) transcription factors (*LOC101210308* and *LOC116401613*) in the salicylic acid (SA) signaling pathway (Figure 7A).

As shown in Figure 7B, in comparison with the control, ETH treatment significantly decreased the IAA content, whereas it enhanced the ABA content. In the auxin signaling transduction pathway, the relative expressions of *auxin transporter-like protein 5* (*CsLAX5*), *CsGH3.17*, and *CsSUAR50* were significantly decreased by ETH treatment (Figure 7C). In ETH signal transduction, the expressions of *CsERS* and *CsERFC3* with ETH treatment were lower than that in the control (Figure 7C). ETH treatment significantly enhanced the expressions of *CsPYL1* and *CsBAK1* (Figure 7C).

## 3. Discussion

As an important small gaseous molecule, ETH affects the growth and development of plants, and it is also a major regulator in response to diverse stresses. Many studies have shown that ETH can induce adventitious rooting in cucumber and marigold [1,15,16]. Using the iTRAQ technique and proteome analysis, Lyu et al. [16] revealed that ETH treatment increased photosynthesis and starch degradation to provide energy for adventitious rooting in cucumber. However, some key genes involved in ETH-promoted adventitious root development in cucumber have not been identified. Here, we revealed the mechanism of ETH-induced adventitious root development at the transcriptional level. First, GO enrichment and KEGG pathways in the transcriptome and proteome were analyzed and compared (Figure 1). In the top 20 KEGG pathway terms, “starch and sucrose metabolism”, “phenylalanine metabolism”, “plant hormone signal transduction”, “fatty acid metabolism”, “pentose and glucuronate interconversions”, “cutin, suberin, and wax biosynthesis”, “photosynthesis–antenna proteins”, “vitamin B6 metabolism”, “purine metabolism”, and “alanine, aspartate, and glutamate metabolism” were enriched in transcriptome and proteome (the order was ranked by transcriptome data), which suggests that exogenous ETH induced adventitious rooting via regulation of the decomposition and metabolism of nutrients, the accumulation and degradation of plant hormones, the synthesis of secondary metabolites, and photosynthesis. In order to identify key genes during ethylene-induced adventitious root development in cucumber, several pathways associated with adventitious rooting and enriched in the KEGG pathway of the transcriptome were selected and analyzed further.

### 3.1. Carbon Metabolism in ETH-Induced Adventitious Root Development

Carbon metabolism is an important metabolic pathway during adventitious root development and can provide metabolites for starch synthesis, a carbon skeleton for amino acid synthesis, and intermediates for secondary metabolism [20]. Further, it influences energy generation and respiration through the TCA and photorespiratory cycles [21].

In our study, 17 DEGs were involved in ETH-inducing adventitious rooting in starch and sucrose metabolism (Figure 2). Among them, HK is considered to be a key enzyme involved in glucose sensing, signal transduction, and photosynthesis [22,23]. Andriunas et al. [24] found that glucose was able to act positively through an HK-dependent signaling pathway, suggesting that glucose could regulate the HK signaling pathway. In our study, ETH treatment decreased the content of glucose, the activity of HK and the relative expression of *CsHK2* during rooting (Figure 2). In *Arabidopsis*, the transcript level of *ERF1* was inhibited by low glucose signaling, which indicated that the ETH might have had an antagonistic effect on low glucose signaling [25,26]. Hence, exogenous ETH might decrease HK activity and *CsHK2* expression during rooting. During this process, energy was obtained by consuming sucrose and glucose. HK enzyme and its related gene in the sucrose metabolism pathway were inhibited by ETH, thus accelerating hypocotyl growth and finally inducing adventitious root development in cucumber. Additionally, starch metabolism may also be involved in adventitious rooting [27]. *β-amylase 1* (*BAM1*) and *β-amylase 3* (*BAM3*) are key genes involved in starch degradation. Starch degradation causes a large amount of starch to be enriched in leaves and delays the growth and development of storage roots [27,28]. Our transcriptome results show that the transcript levels of *CsBAM1* and *CsBAM3* were down-regulated by ETH treatment (Figure 2). Therefore, ETH-treated explants completed energy conversion by reducing the expression of *CsBAM1* and *CsBAM3* in the starch signaling pathway and the content of starch, which may be beneficial for adventitious rooting. According to Lyu et al. [16], 48 h after treatment is the root elongation phase. At this stage, starch might be converted to glucose to provide energy for root growth. Therefore, ETH could induce adventitious rooting by decreasing the transcript levels of the starch degradation-related genes *CsBAM1* and *CsBAM3* and preventing starch degradation.

Further, glycolysis, the TCA cycle, and oxidative phosphorylation are classified as respiratory metabolism and can produce energy in the ATP form [29,30]. In the present study, 12, 5, and 4 DEGs were involved in these pathways by ETH treatment, respectively (Figure 3). A previous study showed that the changes in glycolysis were closely related to the changes in root respiration in sugarbeet (*Beta vulgaris* L.) [29]. The change trend of PK activity was similar to that of the respiratory rate, first increasing, then decreasing, and finally becoming stable [29]. Lyu et al. [16] found that ETH enhanced ATP synthase subunit protein expression at 12 and 24 h, and then declined at 48 h, confirming that ETH-induced adventitious rooting might require energy. This agreed with the findings of our metabolomic analysis showing that the activity of PK and the expression of *CsPK2* were decreased with ETH at 48 h after treatment (Figure 3), suggesting that at this stage, the respiration rate might decrease, resulting in the decline of ATP generation. Hence, ETH could induce adventitious root development by decreasing the PK activity and its transcript level. At 48 h after ETH treatment, ATP synthetase provided more energy for inducing adventitious rooting.

Fatty acids provide energy for some metabolic processes and act as signal transduction regulators [31]. Twelve DEGs were involved in fatty acid biosynthesis and the fatty acid degradation pathway in ETH-induced adventitious root development (Figure 4). Among them, *CYP86A1* was a key gene for root aliphatic suberin biosynthesis in *Arabidopsis* [32,33]. Root suberin could control the absorption and transportation of water and mineral ions from roots to leaves [34]. Therefore, *CYP86A1* also might play an important role in rooting. Our work showed that ETH treatment decreased the expression of *CsCYP86A1* during rooting (Figure 4), suggesting that the energy might be transferred to the hypocotyl to induce adventitious root development.

### 3.2. Secondary Metabolite Biosynthesis Pathway in ETH-Induced Adventitious Root Development

Phenylalanine metabolism and flavonoid biosynthesis are secondary metabolite biosyntheses. They benefit plants throughout the whole developmental process. Wang et al. [35] demonstrated that phenylpropanoid and flavonoid biosynthesis pathways were associated with root development. However, there has been no research showing that phenylalanine metabolism could be involved in adventitious root development. In our study, ETH regulated seven DEGs in phenylalanine metabolism during rooting, including *CsPAL*, *CsPAO*, and *CsAST* (Figure 5). PAL was a critical enzyme that directly catalyzes phenylalanine synthesis and is a precursor for the phenylpropanoid biosynthetic pathway [36]. The activity of PAL varied with the development stage of plants and the differentiation degree of cells and tissues [37,38]. Under excess Fe nutrition, plant growth was suppressed, and the activity of PAL was enhanced [39], indicating that the increase in PAL activity might inhibit plant growth. In our study, the activity and relative expression of PAL decreased with ETH treatment (Figure 5), indicating that the ETH might have induced adventitious rooting by reducing PAL activity and its gene expression.

In addition, flavonoid biosynthesis was reported to have an important role in hairy roots, adventitious roots, and seedling roots [36]. However, there is little information regarding the roles of flavonoid in ETH-induced adventitious rooting. In our transcriptome results, three DEGs, including *CsCHI*, *CsF3′M*, and *CsHCT* were involved in this pathway (Figure 6). Among them, the expression of *CsHCT* was significantly increased by ETH treatment. There was a negative correlation between the expression of *CsHCT* and the content of total flavonoids [40]. When the flavonoid content decreased, the content of lignin increased, and the fibers between the vascular bundles became longer [41]. Hence, ETH treatment might promote adventitious root primordium growth and break through the epidermis by increasing the expression of *CsHCT*, and the newly formed adventitious roots could be connected to the vascular bundle, leading to fiber growth between vascular bundles.

### 3.3. Plant Hormone Signal Transduction in ETH-Induced Adventitious Root Development

Phytohormone is an essential factor affecting adventitious rooting. We identified 26 DEGs in the plant hormone signal transduction pathway in total (Figure 7). Various studies have shown that auxin could induce adventitious root development [42,43]. In the present study, ETH regulated eight DEGs, including *CsAUX1*, *CsAUX/IAA*, *CsGH3*, and *CsSAUR* in the auxin signaling transduction pathway (Figure 7). *AUX1* and *LAX5* are auxin influx carriers of the same family [42]. *AUX1* could mediate the auxin influx specifically in the adventitious root-initiating cells, whereas *LAX5* was involved later than *AUX1* in adventitious rooting [42]. Recently, da Costa et al. [44] reported that *LAX1* and *LAX3* were used as down-regulation genes during adventitious rooting in *Arabidopsis thaliana*, which was consistent with our result. We found that ETH decreased the expression of *CsAUX1* and *CsLAX5* (Figure 7), which proves that the auxin influx genes were negatively regulated in ETH-promoted adventitious root development. Moreover, Harkey et al. [45] found that the ETH precursor ACC reduced the expression of *GH3.17* in *Arabidopsis* roots. The decreased activity of GH3 could lead to a decrease in the IAA level [46]. Our study shows that ETH down-regulated the expression of IAA-amido synthetase *CsGH3.17* (Figure 7). To expand on the results of the expression studies, the endogenous IAA content was further determined in control and ethephon-treated cucumber explants. In our results, ETH significantly decreased the endogenous IAA content (Figure 7). Therefore, ETH could induce adventitious rooting by negatively regulating the auxin signal pathway, in which ETH might decrease the IAA content and the expressions of *CsGH3.17*, *CsAUX1*, and *CsLAX5.*

ABA negatively controls root emergence [47]. However, the crosstalk between ABA and ETH in adventitious root development was unclear. We found that ETH up-regulated the expression of six DEGs in the ABA signal transduction pathway, which included *CsPYR/PYL*, *CsSnRK2*, and *CsABF* (Figure 7). *PYR1*, as a member of the family of PYR/PYL/RCAR ABA receptors, mediated plant immunity, seed germination, and seedling establishment [48,49]. *PYL8* has been reported to induce lateral root growth [50]. Here, we found that ETH up-regulated the expression of the ABA receptors *CsPYR1*, *CsPYL5*, and *CsPYL8* during adventitious rooting (Figure 7), indicating that *CsPYR1*, *CsPYL5*, and *CsPYL8* might positively regulate ETH-induced adventitious root development. Moreover, ETH significantly enhanced ABA content (Figure 7). Thus, ETH could induce rooting by increasing ABA signal transduction-related gene expression and ABA content.

Furthermore, ETH decreased the expression of four DGEs, including *CsETR*, *CsEBF1/2*, and *CsERF1/2* in ETH signal transduction (Figure 7). *ERS* is an ETH receptor gene, which can negatively regulate ETH response [51]. In the present study, ETH decreased the transcript level of *CsERS* (Figure 7). Hua et al. [52] also found that ETH treatment reduced the *ERS* transcript level, indicating that *ERS* could be negatively mediated by exogenous ETH during rooting.

In *Arabidopsis*, exogenous BR could promote adventitious root initiation and growth [53,54]. In our study, ETH up-regulated the expressions of *CsBAK1* and *CsXTH3* in the BR signaling transduction pathway (Figure 7). *BAK1* is a BR receptor gene. Xie et al. [55] reported that graphene oxide could increase *BAK1* expression to enhance root growth in *Brassica napus* L., confirming that *BAK1* might positively respond to root growth. Similarly, our results show that ETH enhanced the transcriptional level of *CsBAK1 (*Figure 7), which suggests that ETH induced adventitious rooting via positively regulating *CsBAK1* expression in BR signal transduction. Therefore, BR might have a positive role in ETH-induced adventitious root development. Xyloglucan endotransglucosylase/hydrolase (XTH), as a cell wall loosening and extension factor, can induce cell elongation and root growth [56]. However, there has been no direct evidence that XTH positively or negatively regulates adventitious root development. We found that ETH up-regulated the expression of *CsXTH3* (Figure 7), which implies that ETH might promote rooting by increasing the transcript level of *CsXTH3*.

## 4. Materials and Methods

### 4.1. Plant Material and Growth Conditions

Cucumber (*C. sativus* L. “Xinchunsihao”) seeds of the same size and full grains were soaked in distilled water for 5–6 h and then placed on a tray with wet filter paper. After this step, the seeds in the tray were moved to the incubator with illumination with a 14 h photoperiod (the photosynthetically active radiation was 200 μmol m^−2^ s^−1^) at 25 ± 1 °C for 7 d, and the filter paper was kept wet in the tray during this period. The primary roots of explants were carefully cut off after 7 d. In accordance with Zhao et al. [57], the induction of adventitious roots was completed 48 h after treatment. Hence, the cucumber explants were placed in Petri dishes with different treatments and allowed to grow for 48 h. The temperature and photoperiod conditions were the same as the above conditions.

### 4.2. Chemicals and Treatments

Based on our previous results [1], 0.5 μM of 2-chloroethylphosphonic acid (ethrel, an ETH donor) was the optimal concentration to induce adventitious rooting in cucumber. After cutting the primary roots, every ten explants of the same size were selected and placed in Petri dishes containing 70 mL of with distilled water or 0.5 μM ethrel, and the solutions used in the study were changed every 24 h [1]. Each experiment was conducted three times with 3 replicates. At 48 h after treatment, samples (cucumber explants) were collected and stored at −80 °C until used.

### 4.3. Gene Expression Profiling Using RNA-Seq

#### 4.3.1. RNA Extraction

The total RNA for RNA-Seq and RT-qPCR analyses was extracted from about 500 mg cucumber explants with Trizol reagent (Sangon, Shanghai, China) in accordance with the manufacturer’s instructions [58]. The total RNA was dissolved in 100 μL of RNase-free water and quantified using a NanoDrop spectrophotometer (Thermo Scientific, Waltham, MA, USA) [59]. The RNA quality was evaluated using the 6000 Pico LabChip of the Agilent 2100 Bioanalyzer (Agilent, Shanghai, China) [60]. RNA samples were sent to Novogene Co., Ltd. (Beijing, China) for cDNA library construction and sequencing. The reference genome for transcriptome analysis was acquired from https://www.ncbi.nlm.nih.gov/genome/1639?genome_assembly_id=749658 (accessed on 18 March 2020). Each treatment had 3 technical replications and 3 biological replications.

#### 4.3.2. cDNA Synthesis and Illumina Sequencing

The mRNA was purified from the total RNA using poly-T oligo-attached magnetic beads [61]. In the NEBNext First Strand Synthesis Reaction Buffer (5×), divalent cations were used for fragmentation at a high temperature. First-strand cDNA was synthesized using random hexamer primer and M-MuLV Reverse Transcriptase (RNase H-). Second-strand cDNA synthesis was subsequently performed using DNA Polymerase I and RNase H. Remaining overhangs were converted into blunt ends via exonuclease/polymerase activities [62]. After adenylation of 3′ ends of the DNA fragments, a NEBNext Adaptor with a hairpin loop structure was ligated to prepare for hybridization. In order to screen 250–300 bp cDNA fragments preferentially, the library fragments were purified using an AMPure XP system (Bethe controlman Coulter, Beverly, CA, USA) [20]. Before PCR, 3 µL of USER Enzyme (NEB, Boston, MA, USA) was used at 37 °C for 15 min followed by 5 min at 95 °C with size-selected, adaptor-ligated cDNA [63]. Then, PCR was performed with Phusion High-Fidelity DNA polymerase, Universal PCR primers, and Index (X) Primer. At last, the PCR products were purified (AMPure XP system; Bethe controlman Coulter, Beverly, CA, USA) and the library quality was assessed on the Agilent Bioanalyzer 2100 system (Agilent, Shanghai, China) [20]. The library products were sequenced with the Illumina Novaseq platform. The RNA-seq data produced in this study were deposited in the NCBI Gene Expression Omnibus (GSE189315).

#### 4.3.3. Bioinformatics Analysis

The raw reads were raw sequencing data, and they could be transferred from the original image data through a base call. The clean reads were a high-quality sequence after clearing data from the original reads. The error rate was the total sequencing error rate of the data. Q20 and Q30 refer to the percentage of bases with Phred values greater than 20 and 30 to the total bases, respectively. GCpct was the percentage of G and C among the four bases in the clean reads. The transcriptome sequencing was of high quality and can be used for further analysis (Appendix A). In addition, the Pearson correlation analysis showed high consistency of gene expression patterns of different biological replicates of each sample, suggesting the high reproducibility of transcriptome data (Appendix A). According to the RPKM (reads per kilobase per million reads) criteria, levels of DEGs were calculated by quantifying reads. Transcriptome libraries should be based on the criteria: |log_2_(FoldChange)| > 1 and padj ≤ 0.05. Analyses of GO functional enrichment and KEGG pathway enrichment of the DEGs were carried out using ClusterProfiler software. In the functional enrichment analysis of GO, the DEGs were mainly enriched in the biological process (BP), molecular function (MF), and cellular component (CC). Using the KAAS server, the metabolic pathway was analyzed according to the KEGG pathway database (http://www.genome.jp/kegg/pathway.html) (accessed on 25 May 2020) [64].

Additionally, because the treatment time and ETH donor ethrel concentration were the same in our transcriptome analysis and in the proteomic analysis of Lyu et al. [16], we compared the DEGs in the transcriptome with differentially expressed proteins (DEPs) in the proteome.

### 4.4. Determination of Sucrose, Glucose, and Starch Content

The determination of sucrose content was performed according to Tauzi and Giardina [65]. On the basis of our experimental requirements, we modified the procedure appropriately. The cucumber explants (1.0 g) were added to 10 mL of 80% ethanol and diluted with distilled water in a 100 mL volumetric flask. The volumetric flask was placed in a water bath at a constant temperature of 80 °C for 45 min, then cooled to room temperature, and filtered with filter paper. This step was repeated twice, and the filtrate was collected as the reaction solution. The filtrate (0.4 mL) was mixed with 0.2 mL of 2 M sodium hydroxide (NaOH), and then bathed at 80 °C for 10 min. After cooling to room temperature, the sucrose content was determined. The absorbance was measured at 540 nm, and the soluble sugar content was calculated according to the standard curve.

The determination of the glucose content was performed according to Smith and Zeeman [66]. Fresh explant (0.5 g) was homogenized with 10 mL distilled water, diluted with distilled water, and mixed in a 50 mL volumetric flask. Then it was placed in a boiling water bath for 10 min. The reaction mixture contained 3,5-dinitrosalicylic acid (DNS) regent (6.3 g), 262 mL 2 M NaOH solution, 185 g potassium sodium tartrate with 500 mL of hot water, 5 g of redistilled phenol, and 5 g of sodium sulfite. After cooling the mixture to room temperature, the volume was accurately adjusted to 1000 mL with distilled water. Then the homogenate was centrifuged at 4000 rpm for 15 min at 4 ℃. The supernatant (2 mL) and deoxyribonucleic acid (1.5 mL) were mixed together, and the mixture was centrifuged at 4000 rpm for 15 min at 4 °C. Then it was placed in boiling water at 100 °C and bathed for 5 min. After cooling to room temperature, the solution was filtered into a 25 mL volumetric flask. The sample absorbance was measured at 540 nm against the controls.

For the starch content analysis, a 0.5 g sample was homogenized with 2 mL of distilled water and 3.2 mL of 60% HClO_4_. Distilled water was added to the mixture to 100 mL and then centrifuged at 5000 rpm for 5 min. Then, 0.5 mL of the supernatant was pipetted and diluted to 3 mL with distilled water. Then, 2 mL of iodine reagent was added, mixed well, and the mixture was left standing for 5 min. Finally, the solution was added to 10 mL of distilled water. The optical density (OD) was monitored at 660 nm [66].

### 4.5. Enzyme Assays

#### 4.5.1. Determination of Sucrose Synthase (SS), Sucrose–Phosphate Synthase (SPS), and Hexokinase (HK) Activities

The enzyme activities of SS, SPS, and HK were determined by enzyme-linked immunosorbent assay (ELISA; Andy Gene Biotechnology Co., Ltd., Beijing, China). A 1.0 g sample was ground with liquid nitrogen and homogenized with 9 mL PBS (pH = 7.4). Then the homogenate was centrifuged at 3000 rpm for 15 min at 4 °C [67]. The subsequent measurement method was carried out on the basis of the manufacturer’s instructions.

#### 4.5.2. Determination of Phenylalanine Ammonia-Lyase (PAL) Activity

The PAL activity was assessed according to González-Mendoza et al. [68]. A fresh sample (0.5 g) was mixed into 5 mL of Tris-HCl buffer (0.1 M, pH 7.5), 1 mM 1% polyvinylpolypyrrolidone and ethylenediaminetetraacetic acid (EDTA). Then the mixture was centrifuged at 12,000 rpm, at 4 °C for 10 min. The reaction mixture, including 100 µL of enzyme extract and 200 µL of Tris-HCl (50 mM, pH 8.8) containing 20 mM L-phenylalanine was incubated at 40 °C for 30 min. Then 100 µL of 1 M hydrochloric acid (HCl) was added to stop the reaction. The absorbance value was measured at 290 nm. The amount of enzyme that increased the OD_290_ by 0.01 per minute was defined as an enzyme activity unit IU.

### 4.6. Determination of IAA and ABA Contents

The concentrations of the IAA and ABA contents were determined by Quaternary gradient ultra-fast liquid chromatograph using a Waters Acquity ARC 600-2998 equipped with the Symmetry-C18 column (4.6 × 250 mm, 5 μm). The frozen sample (1.0 g) was taken and ground with liquid nitrogen quickly and then transferred to a 10 mL centrifuge tube. Then, 5 mL of 80% cold pure methanol was added and the sample was stored in the dark in the refrigerator at 4 °C for 12 h. After 12 h, the sample was centrifuged at a speed of 4000 rpm for 15 min at 4 °C. Then the supernatant was pipetted into another 10 mL centrifuge tube. The residues were suspended in 2.5 mL of extraction solution and stored at 4 °C for 1 h. Then the mixture was centrifuged at a speed of 4000 rpm for 15 min, and the filtrate was collected and combined in the tube. This step was repeated twice. The filtrate was diluted to 10 mL with 80% methanol. Two mL of the supernatant was evaporated in a rotary vacuum at 38 °C for 4 h and then dissolved with 2 mL of 80% cold pure methanol. The extract was filtered with a 0.22 μm filter for liquid chromatograph detection. The mobile phase was chromatographic methanol (A) + 0.1% phosphoric acid (B). The flow rate was 1.0 mL min^−1^. The wavelength was 254 nm, and the column temperature was 30 °C [69].

### 4.7. RT-qPCR Assays

In accordance with the manufacturer’s instructions, the RNA (400 ng) was reverse-transcribed with 5 × *Evo M-MLVRT* Master Mix (AG, Changsha, China). SYBR Green Pro Taq HS Premix (AG, Changsha, China) was used to amplify cDNA. The gene expression data were normalized using *Csactin* as an internal control [70]. The primers were designed by primerbank (https://pga.mgh.harvard.edu/primerbank/) (accessed on 1 May 2021). The RT-qPCR reactions containing 7.2 μL of ddH_2_O, 0.4 μL of forward primer, 0.4 μL of reverse primer, 2 μL of cDNA (800 ng), and 10 μL of *Taq* enzyme were performed on the ABI StepOne plus system (Foster, CA, USA) with the following program: 95 °C for 30 s, 40 cycles of 95 °C for 5 s and 60 °C for 30 s. In order to assess possible contaminations and primer dimer formations, no-template controls (NTCs) were used as a negative control [71]. The RT-qPCR efficiencies of amplicons were calculated by performing a standard curve. Moreover, melting curve analysis was conducted to ensure the amplification of specific amplicons. StepOne^TM^ Software v2.3 (Life Technologies, Boston, MA, USA) was used to analyze the ct values, and 2^−ΔΔct^ was used to calculate the normalized gene expression levels [72]. The experiment was conducted with three bio-replicates. The primers are listed in Appendix A.

### 4.8. Statistical Analysis

The statistical analysis was conducted using the Statistical Package for Social Sciences 22.0 (SPSS 22.0, Inc., Chicago, IL, USA) software. All results were expressed as the mean values ± standard error (SE) from at least three independent duplications. The data from substance contents, enzyme activities, plant hormone contents and RT-qPCR assays were analyzed by *t*-test at *p* < 0.01 and *p* < 0.05, respectively.

## 5. Conclusions

In conclusion, the candidate genes of sucrose and starch metabolism (*CsHK2*), glycolysis-related gene (*CsPK2*), phenylalanine metabolism (*CsPAL*), flavonoid biosynthesis (*CsF3′M*), auxin signaling transduction (*CsLAX5*, *CsGH3.17*, and *CsSUAR50*), and ETH signaling transduction (*CsERS*) were ETH-down-regulated genes, whereas the candidate genes of starch metabolism (*CsBAM1* and *CsBAM3*), phenylalanine metabolism (*CsPAO*), flavonoid biosynthesis (*CsHCT*), ABA signaling transduction (*CsPYL1*, *CsPYL5*, and *CsPYL8*), and BR signaling transduction (*CsBAK1* and *CsXTH3*) were up-regulated in ETH-induced adventitious rooting. These candidate genes were associated with ETH-induced adventitious root development. This study will be useful for understanding the molecular mechanisms of ETH-inducing adventitious rooting.

## Figures and Tables

**Figure 1 ijms-23-12981-f001:**
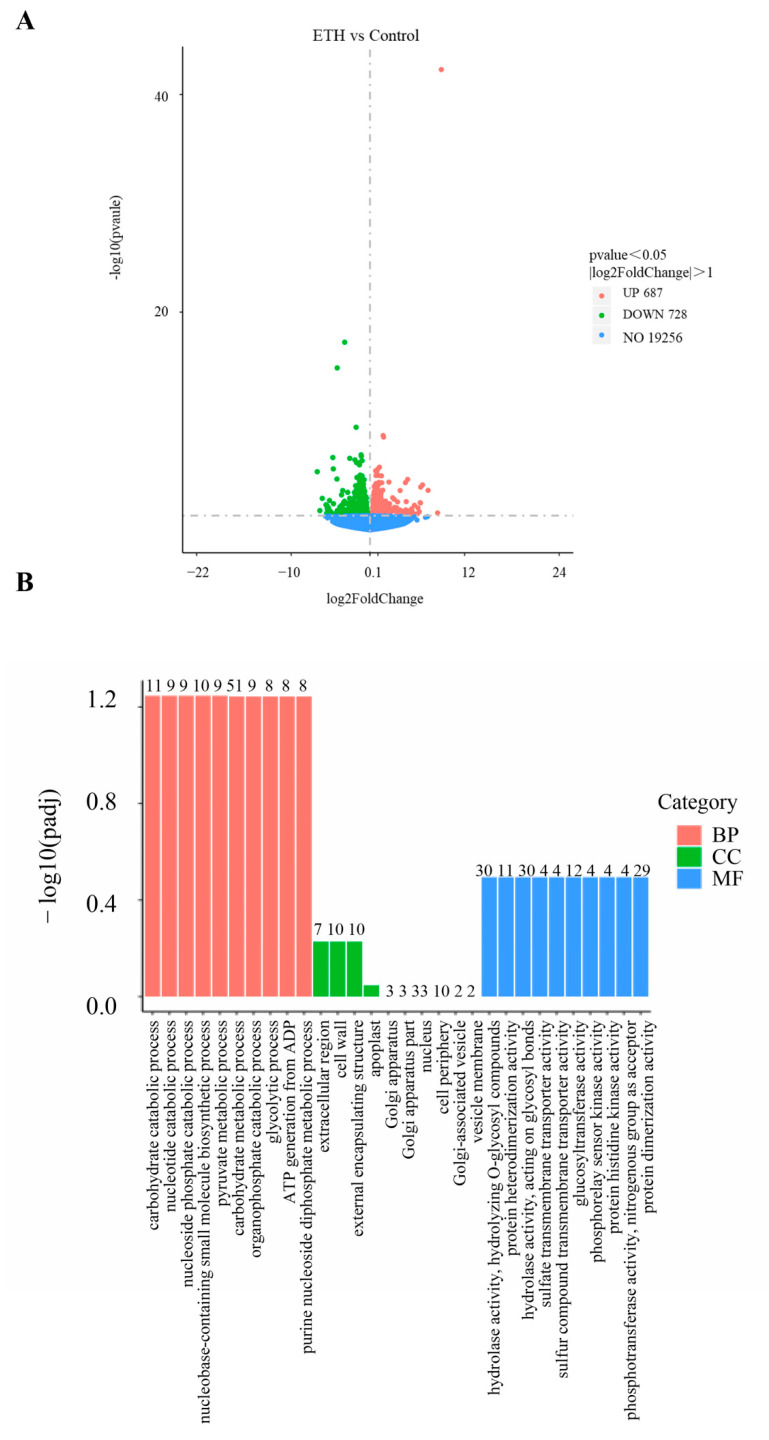
Volcano map, GO classification and KEGG pathway of DEGs identified in cucumber by ETH treatment. (**A**) Volcano map of DEGs. The X-axis displays the log2FoldChange value, and the Y-axis displays the −log10p value. Red and blue dots represent the number of up- and down-regulated DEGs, respectively. (**B**) GO classification of the annotated DEGs. The red, green and blue bars represent biological process (BP), cell component (CC) and molecular function (MF), respectively. The Y-axis displays the −log10 (padj) value of each term, and the X-axis displays the GO terms and the enriched DEGs numbers.

**Figure 2 ijms-23-12981-f002:**
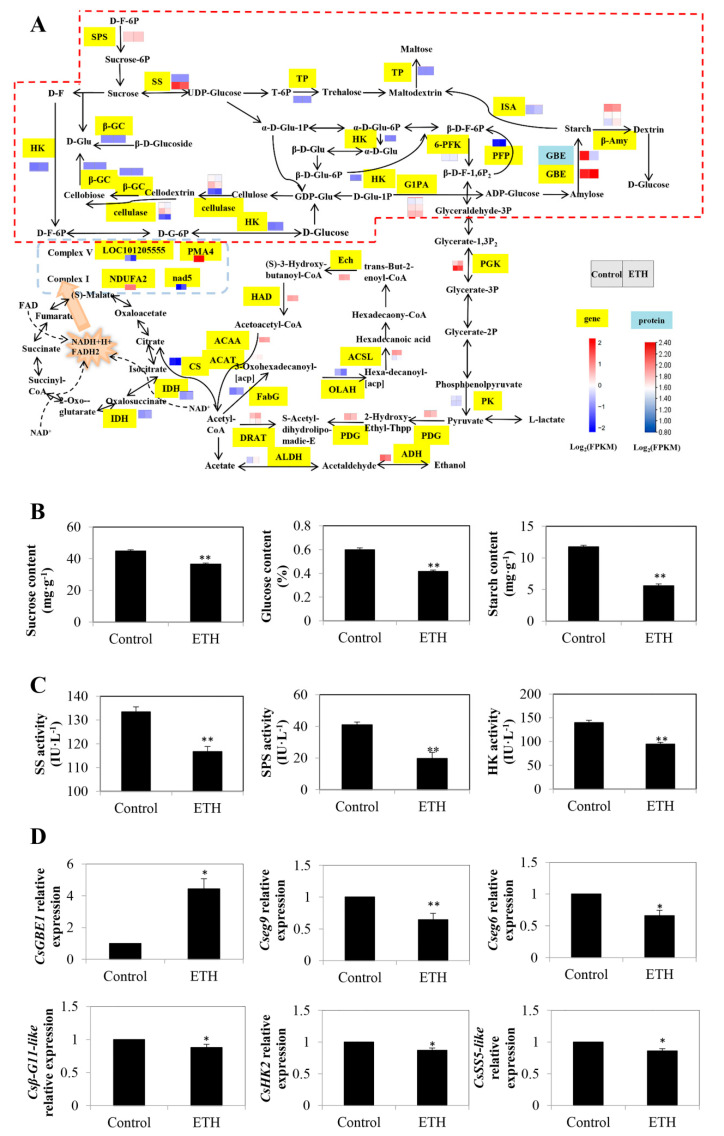
Identification and validation of key DEGs in starch and sucrose metabolism during adventitious root development in cucumber. (**A**) Expression patterns of genes and proteins for starch and sucrose metabolism in carbon metabolism during adventitious root development in cucumber. The dashed box represents starch and sucrose metabolism. Annotations of genes and proteins are available in Appendix A. Black characters with yellow background are genes, whereas black characters with blue background are proteins. The two squares under the names of genes or proteins indicate the abundance change between the control and ETH. A red square means up-regulation, whereas a blue square means down-regulation. (**B**) The contents of sucrose, glucose and starch during adventitious root development in cucumber. (**C**) The activities of SS, SPS and HK enzyme during adventitious root development in cucumber. (**D**) Quantitative real-time PCR analysis of the expression profiles of 6 genes related to starch and sucrose metabolism during adventitious root development in cucumber. Asterisk (*) and double asterisk (**) indicate the significance at *p* < 0.05 and *p* < 0.01, respectively. The bar represents the value of standard error. Values represent the mean ± SE of three replicates. Each experiment was repeated three times independently.

**Figure 3 ijms-23-12981-f003:**
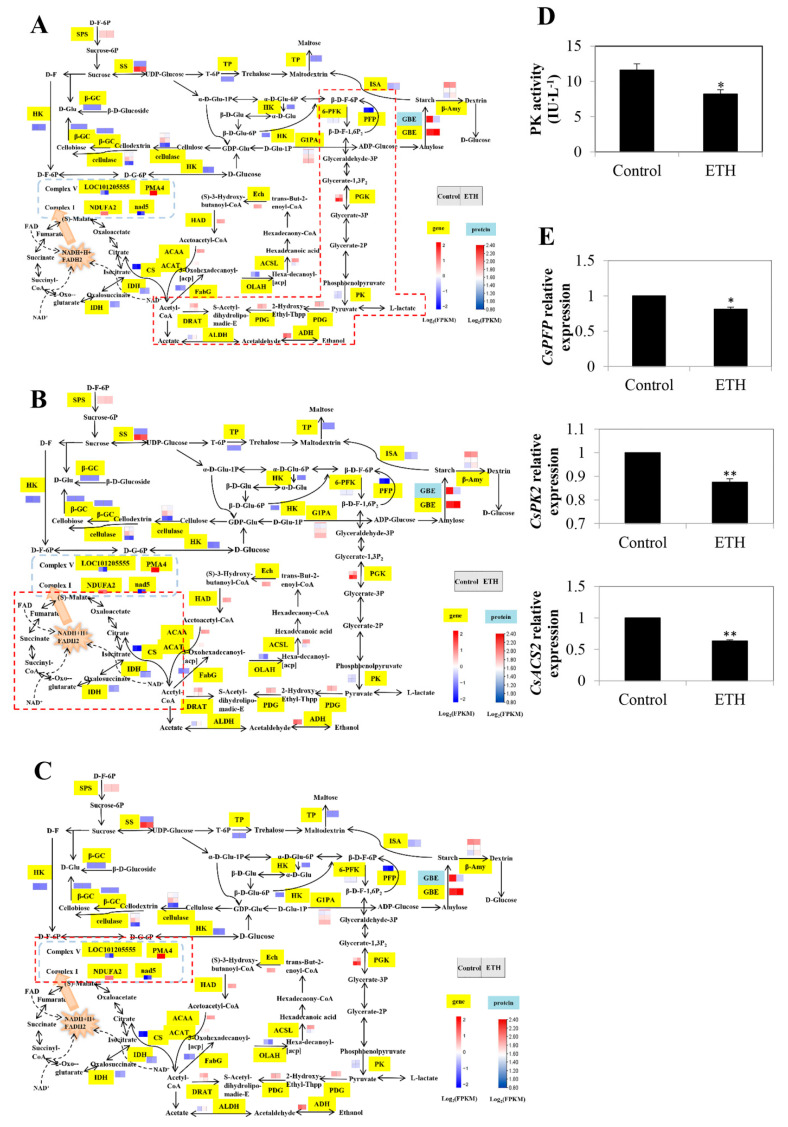
Identification and validation of key DEGs in glycolysis/gluconeogenesis, citrate cycle (TCA cycle) and oxidative phosphorylation during adventitious root development in cucumber. (**A**) Expression patterns of genes for glycolysis/gluconeogenesis during adventitious root development in cucumber. (**B**) Expression patterns of genes for TCA cycle in carbon metabolism during adventitious root development in cucumber. (**C**) Expression patterns of genes for oxidative phosphorylation in carbon metabolism during adventitious root development in cucumber. Annotations of genes and proteins are available in Appendix A. Black characters with yellow background are genes. The two squares under the names of genes or proteins indicate the abundance change of the control and ETH. A red square means up-regulation, whereas a blue square means down-regulation. The red dotted box represents the different pathways involved in carbon metabolism. (**D**) The activity of PK enzyme during adventitious root development in cucumber. (**E**) Quantitative real-time PCR analysis of the expression profiles of 6 genes related to glycolysis/gluconeogenesis, TCA cycle and oxidative phosphorylation during adventitious root development in cucumber. Asterisk (*) and double asterisk (**) indicate the significance at *p* < 0.05 and *p* < 0.01, respectively. The bar represents the value of standard error. Values represent the mean ± SE of three replicates. Each experiment was repeated three times independently.

**Figure 4 ijms-23-12981-f004:**
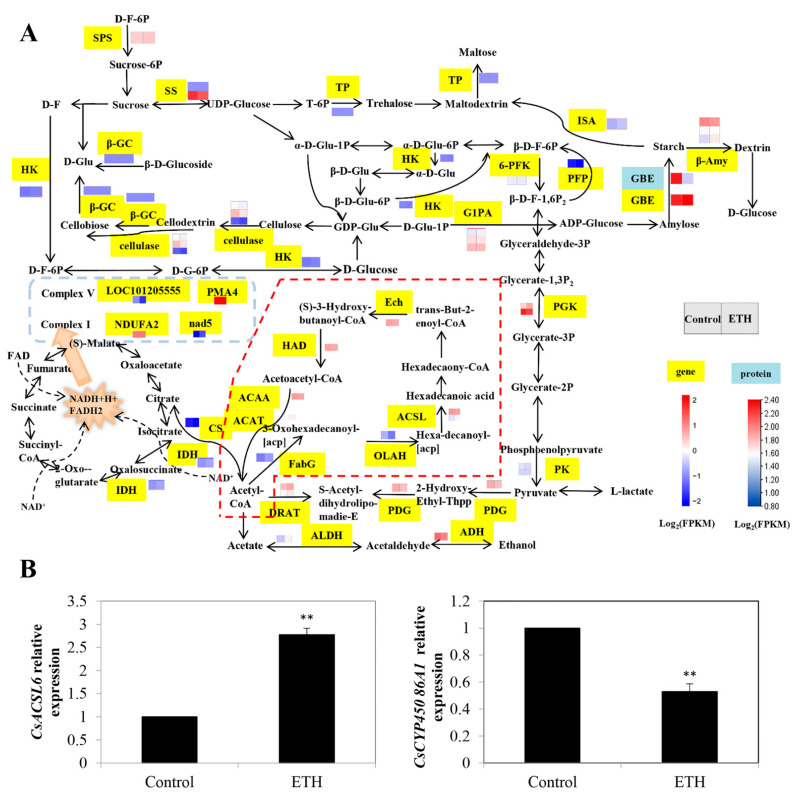
Identification and validation of key DEGs in fatty acid biosynthesis and fatty acid degradation during adventitious root development in cucumber. (**A**) Expression patterns of genes and proteins for fatty acid biosynthesis and fatty acid degradation in carbon metabolism during adventitious root development in cucumber. The dashed box represents fatty acid biosynthesis and fatty acid degradation. Annotations of genes and proteins are available in Appendix A. Black characters with yellow background are genes, whereas black characters with blue background are proteins. The two squares under the names of genes or proteins indicate the change in abundance of the control and ETH. A red square means up-regulation, whereas a blue square means down-regulation. (**B**) The expression profiles of 2 genes related to fatty acid biosynthesis and fatty acid degradation during adventitious root development in cucumber. Double asterisk (**) indicates the significance at *p* < 0.01. The bar represents the value of standard error. Values represent the mean ± SE of three replicates. Each experiment was repeated three times independently.

**Figure 5 ijms-23-12981-f005:**
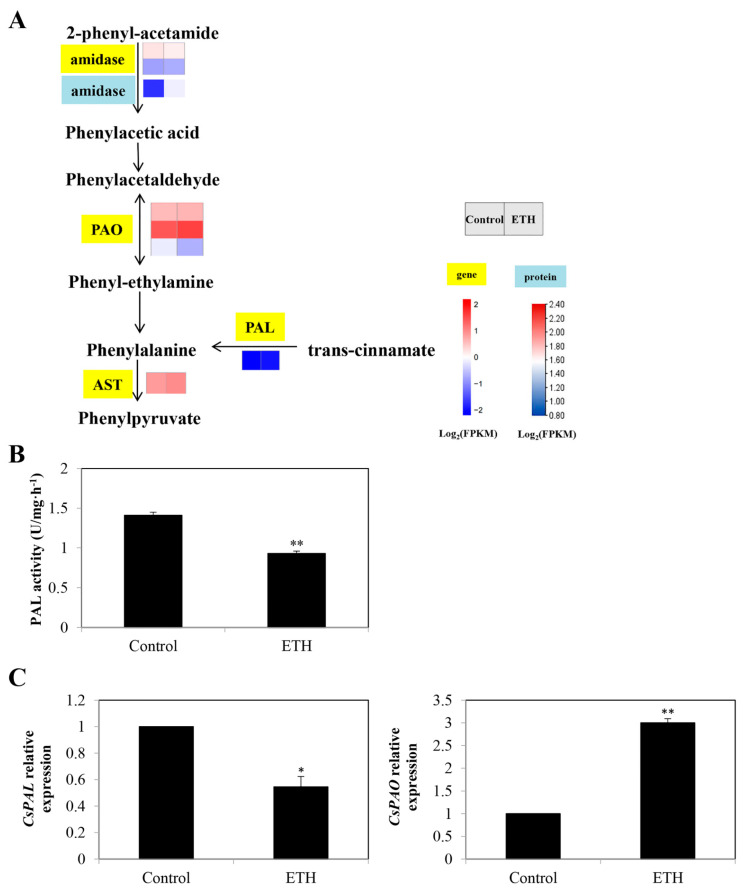
Identification and validation of key DEGs in phenylalanine metabolism during adventitious root development in cucumber. (**A**) Expression patterns of genes and proteins for phenylalanine metabolism in secondary metabolism during adventitious root development in cucumber. Annotations of genes and proteins are available in Appendix A. Black characters with yellow background are genes, whereas black characters with blue background are proteins. The two squares under the names of genes or proteins indicate the change in abundance of the control and ETH. A red square means up-regulation, whereas a blue square means down-regulation. (**B**) The activity of PAL and PAO enzyme during adventitious root development in cucumber. (**C**) Quantitative real-time PCR analysis of the expression profiles of 2 genes related to phenylalanine metabolism during adventitious root development in cucumber. Asterisk (*) and double asterisk (**) indicate the significance at *p* < 0.05 and *p* < 0.01, respectively. The bar represents the value of standard error. Values represent the mean ± SE of three replicates. Each experiment was repeated three times independently.

**Figure 6 ijms-23-12981-f006:**
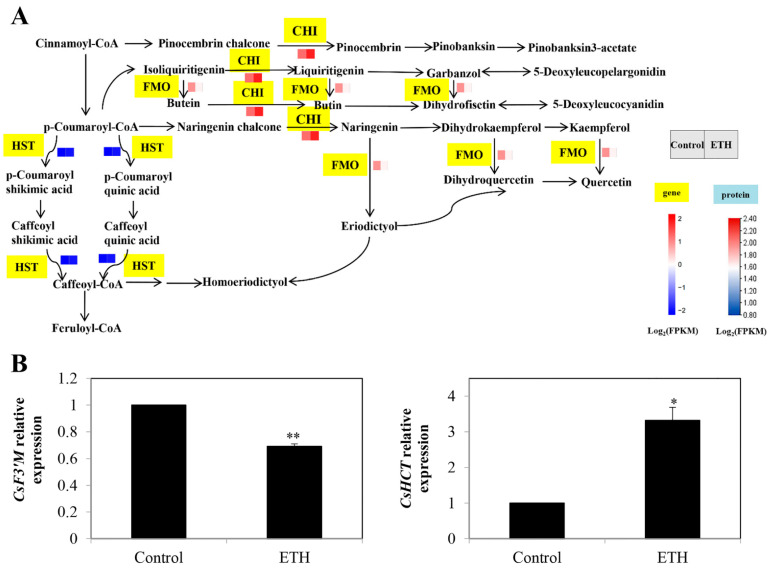
Identification and validation of key DEGs in flavonoid biosynthesis during adventitious root development in cucumber. (**A**) Expression patterns of genes and proteins for flavonoid biosynthesis in secondary metabolism during adventitious root development in cucumber. Annotations of genes and proteins are available in Appendix A. Black characters with yellow background are genes. The two squares under the names of genes or proteins indicate the change in abundance of the control and ETH. A red square means up-regulation, whereas a blue square means down-regulation. (**B**) The expression profiles of 2 genes related to flavonoid biosynthesis during adventitious root development in cucumber. Asterisk (*) and double asterisk (**) indicate the significance at *p* < 0.05 and *p* < 0.01, respectively. The bar represents the value of standard error. Values represent the mean ± SE of three replicates. Each experiment was repeated three times independently.

**Figure 7 ijms-23-12981-f007:**
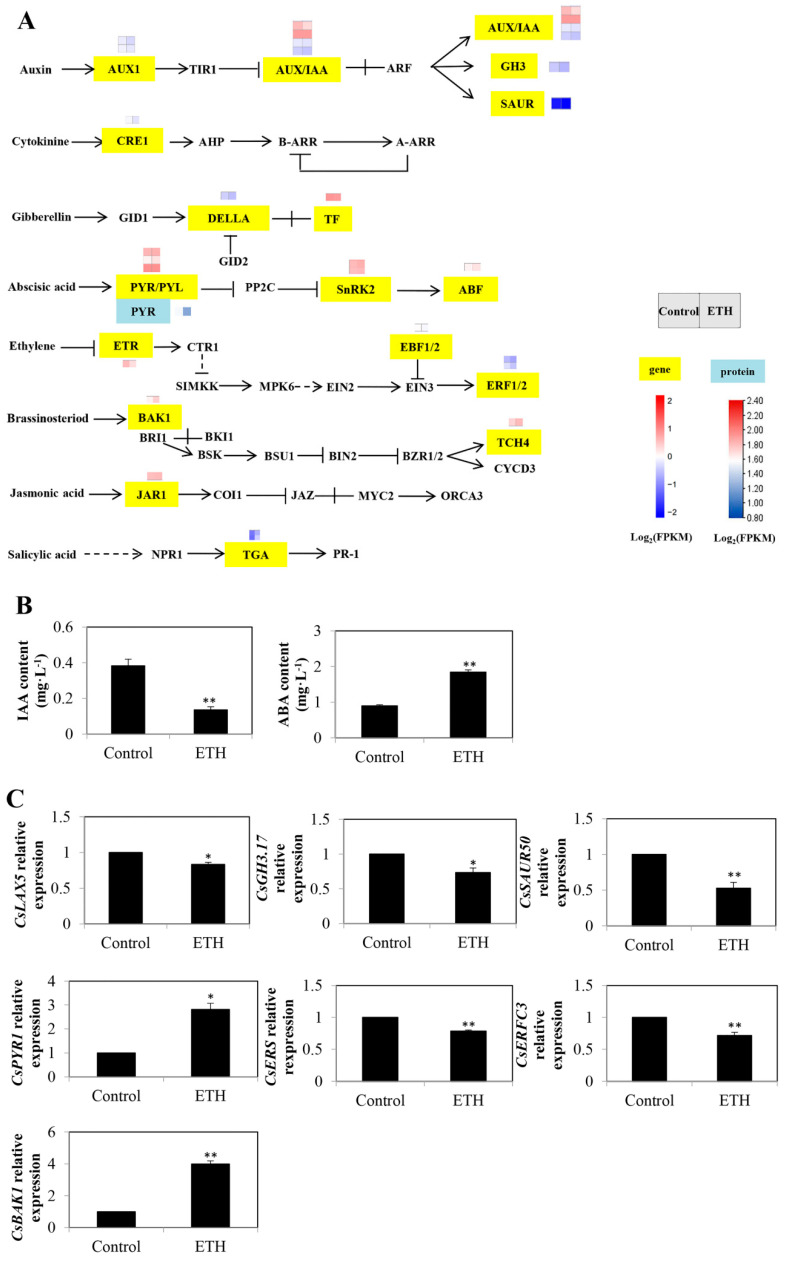
Identification and validation of key DEGs in plant hormone signal transduction during adventitious root development in cucumber. (**A**) Expression patterns of genes and proteins for plant hormone signal transduction during adventitious root development in cucumber. Annotations of genes and proteins are available in Appendix A. Black characters with yellow background are genes, whereas black characters with blue background are proteins. The two squares under the names of genes or proteins indicate the change in abundance of the control and ETH. A red square means up-regulation, whereas a blue square means down-regulation. (**B**) The contents of IAA and ABA during adventitious root development in cucumber. (**C**) Quantitative real-time PCR analysis of the expression profiles of 7 genes related to plant hormone signal transduction during adventitious root development in cucumber. Asterisk (*) and double asterisk (**) indicate the significance at *p* < 0.05 and *p* < 0.01, respectively. The bar represents the value of standard error. Values represent the mean ± SE of three replicates. Each experiment was repeated three times independently.

**Table 1 ijms-23-12981-t001:** Enriched KEGG pathways of the DEGs in ETH treatment.

KEGGID	Pathway	*p* Value	Total DEGs	Up-Regulated DEGs	Down-Regulated DEGs
csv00591	Linoleic acid metabolism	0.006637889	6	4	2
csv00500	Starch and sucrose metabolism	0.007112864	17	2	15
csv00909	Sesquiterpenoid and triterpenoid biosynthesis	0.007603753	5	2	3
csv00071	Fatty acid degradation	0.017687328	8	6	2
csv00950	Isoquinoline alkaloid biosynthesis	0.018821384	4	4	0
csv00360	Phenylalanine metabolism	0.021066898	7	5	2
csv04075	Plant hormone signal transduction	0.026635092	26	13	13
csv01212	Fatty acid metabolism	0.032091296	9	7	2
csv00040	Pentose and glucuronate interconversions	0.037498837	10	2	8
csv00073	Cutin, suberine and wax biosynthesis	0.039280085	5	0	5

## Data Availability

Data are contained within the article or Appendix A.

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
