# Peer review of "Identification of Key Genes during Ethylene-Induced Adventitious Root Development in Cucumber (Cucumis sativus L.)"

_ijms, 2022, doi:10.3390/ijms232112981_

Round 1

Reviewer 1 Report (Previous Reviewer 2)

Authors improved the original manuscript and I suggest the publication of resubmitted version in the present form.

Author Response

Response to Reviewer 1 Comments

Point 1: Authors improved the original manuscript and I suggest the publication of resubmitted version in the present form.

Response 1: Thank you very much for your comments. We are pleased to have been given the opportunity to publish our manuscript.

Reviewer 2 Report (New Reviewer)

Comments to the authors:

The paper entitled “Identification of key genes during ethylene-induced adventitious root development in cucumber (Cucumis sativus L.)” is an original research article. The paper will contribute to the science as well. The article has novelty, and the methods are modern.

The topic selection and research methods of this article are relatively excellent. The author has also done a lot of literature, the research content is relatively rich, and the relevant data are also relatively detailed.

1.     The abstract should emphasize the key points and strengthen the logic between the research background, content, results, analysis, and discussion, rather than listing all the results. Moreover, the highlights and key points of this work should be highlighted in the abstract.

2.     The keywords need to be different from the words used in the title of the manuscript.

3.     The introduction needs to focus on the literature related to the topic. The point lack in this section is connectivity between the paragraphs and sentences.

4.     The methodology overall is not in an organized form, need to improve the presentation of this section.

5.     The comparative discussion needs to be stronger. Needs more evidence from the previous literature to compare with the current results obtained.  

6.     The sentences in the text should be scientific and standardized and should be carefully checked in the overall manuscript (for example, please refer to other literature for the method section).

7.     The figures’ resolution needs to be improved and clear, currently, all the figures are not in good resolution.

8.     Since it is an English paper, the sentences should be smooth and reasonable, and it is best to find an English-speaking expert to check.

Author Response

Response to Reviewer 2 Comments

Point 1: The paper entitled “Identification of key genes during ethylene-induced adventitious root development in cucumber (Cucumis sativus L.)” is an original research article. The paper will contribute to the science as well. The article has novelty, and the methods are modern.

The topic selection and research methods of this article are relatively excellent. The author has also done a lot of literature, the research content is relatively rich, and the relevant data are also relatively detailed.

Response 1: Thanks very much for your attention and accurate evaluation on our paper again. We know that your suggestions are a great help to us to improve our paper structure. We hope you will be satisfied with our efforts. More details please see the revised manuscript.

Point 2: The abstract should emphasize the key points and strengthen the logic between the research background, content, results, analysis, and discussion, rather than listing all the results. Moreover, the highlights and key points of this work should be highlighted in the abstract.

Response 2: Thank you for the valuable comment. As you suggested, We added the background and highlights of our study in the abstract in the revised manuscript as follows.

“Ethylene (ETH), as a key plant hormone, plays critical roles in various processes of plant growth and development. ETH has been reported to induce adventitious rooting. Moreover, ….” was added in Page 1 Lines 11-12.

“This work identified the key pathways and genes in ETH-induced adventitious rooting in cucumber, which …” was added in Please see Page 1 Lines 34-35.

Point 3: The keywords need to be different from the words used in the title of the manuscript.

Response 3: We really appreciate for contributing such a good idea to the improvement of this manuscript. We removed some unnecessary keywords in the revised manuscript.

“Keywords: ethylene; transcriptome; adventitious root development; carbon metabolism; secondary metabolism; plant hormone signal transduction” was changed to “Keywords: transcriptome; carbon metabolism; secondary metabolism; plant hormone signal transduction“ in Page 1 Lines 38-39.

Point 4: The introduction needs to focus on the literature related to the topic. The point lack in this section is connectivity between the paragraphs and sentences.

Response 4: Thank you for your kind suggestion. This is such a valuable idea to improve our manuscript. We added some sentences in the introduction section in the following.

“Although ETH may influence many aspects of plant growth and development, the roles of ETH in adventitious root development are still unclear.” was added in Page 2 Lines 58-59.

“ETH and its crosstalk with other signaling molecules could regulate adventitious root development. However, many studies have only elucidated its mechanism from the perspective of pharmacology, and the genes or proteins in ETH-induced adventitious rooting are still unknown.” was added in Page 2 Lines 79-82.

Point 5: The methodology overall is not in an organized form, need to improve the presentation of this section.

Response 5: Thank you very much for your kind comment. According to your comment, we added some information of Materials and Methods in the following.

“The ETH donor was 2-chloroethylphosphonic acid (ethrel). Based on our previous results [1], 0.5 μM of ethrel was …” was changed to “Based on our previous results [1], 0.5 μM of 2-chloroethylphosphonic acid (ethrel, an ETH donor) was …” in Page 21 Lines 529-530.

“… and the solutions were changed every 24 h. ” was changed to “… and the solutions used in the study were changed every 24 h [1].” in Page 21 Line 533.

“Each experiment was conducted three times with 3 replicates.” was added in Page 21 Lines 533-534.

“(…, Waltham, MA, USA)” was added in Page 21 Line 542.

“The reference genome for transcriptome analysis was acquired from https://www.ncbi.nlm.nih.gov/genome/1639?genome_assembly_id=749658.” was added in Page 21 Lines 545-546,

“In addition, the Pearson correlation analysis showed high consistency of gene expression patterns of different biological replicates of each sample, suggesting the high reproducibility of transcriptome data (Supplementary Figure S1). According to the RPKM (reads per kilobase per million reads) criteria, levels of DEGs were calculated by quantifying reads.”was revised and added in Page 22 Lines 575-579.

“Moreover, we used |log2(FoldChange)| > 1 and padj ≤ 0.05 as the threshold to judge the significance of the gene expression difference between the control and ETH treatment.” was changed to “Transcriptome libraries should be based on the criteria: |log2(FoldChange)| > 1 and padj ≤ 0.05 .” in Page 22 Lines 580-581.

“ClusterProfiler software was used for GO functional enrichment analysis and KEGG pathway enrichment analysis of the DEGs.” was changed to “Analysis of GO functional enrichment and KEGG pathway enrichment of the DEGs were carried out using ClusterProfiler software.” in Page 22 Lines 581-582.

Point 6: The comparative discussion needs to be stronger. Needs more evidence from the previous literature to compare with the current results obtained.

Response 6: Thank you for this suggestion. We agree with your insightful comment. We have carefully checked the discussion section and added more evidence from the related literature to compare with the current results obtained. Thus, we improved some statements in our revised draft. The related statements are as follows.

“Andriunas et al. [24] found that glucose was able to act through an HK-dependent signaling pathway.” was changed to “… act positively through an HK-dependent signaling pathway, suggesting that glucose could regulate the HK signaling pathway” in Pages 17 Lines 386-388.

“In our study, ETH treatment decreased the activity of HK and the relative expression of CsHK2 during rooting (Figures 2 and 3).” was changed to “… decreased the content of glucose, the activity of HK and …(Figure 2).” in Page 17 Lines 388-389.

The word “low” was added in Page 17 Line 391.

“During this process, energy was obtained by consuming sucrose and glucose. HK enzyme and its related gene in sucrose metabolism pathway were inhibited by ETH, thus accelerating hypocotyl growth and finally inducing adventitious root development in cucumber.” was added in Pages 17-18 Lines 393-396.

“Therefore, ETH-treated explants completed energy conversion by reducing the expression of CsBAM1 and CsBAM3 in starch signaling pathway and the content of starch, which may be beneficial for adventitious rooting.” was added in Page 18 Lines 401-404.

“In our work, the activity of PK and the expression of CsPK2 were decreased with ETH (Figure 3), …”was changed to “This agreed with the findings of our metabolomic analysis showing that the activity of PK and the expression of CsPK2 were decreased with ETH at 48 h after treatment (Figure 3), …” in Page 18 Lines 417-419.

“In addition, flavonoid biosynthesis was reported to have an important role in adventitious root development [36].” was changed to “… have an important role in hairy roots, adventitious roots, and seedling roots [36].” in Page 19 Lines 449-450.

“However, there is little information regarding the roles of flavonoid of ETH-induced adventitious rooting.” was added in Page 19 Lines 450-451.

“…, , which was consistent with our result.” was added in Page 19 Line 471.

Point 7: The sentences in the text should be scientific and standardized and should be carefully checked in the overall manuscript (for example, please refer to other literature for the method section).

Response 7: We appreciate the reviewer’s perspective. We revised some sentences in accordance with your suggestion. Related statements in the revision are as follows.

The word “then” was added in Page 21 Line 519.

“After this step,” was added in Page 21 Lines 519-520.

The word “is” was changed to “was” in Page 22 Line 573.

“All results are expressed as the mean values ± standard error (SE) from at least three independent duplications. The statistical analysis was conducted using the Statistical Package for Social Sciences 22.0 (SPSS 22.0, Inc., Chicago, IL, USA) software. The different treatments were separated by a t-test at P<0.01 and P<0.05.” was changed to “The statistical analysis was conducted using the Statistical Package for Social Sciences 22.0 (SPSS 22.0, Inc., Chicago, IL, USA) software. All results were expressed as the mean values ± standard error (SE) from at least three independent duplications. The data from substance contents, enzyme activities, plant hormone contents and RT-qPCR assays were analyzed by t-test at P<0.01 and P<0.05, respectively.” in Page 24 Lines 676-680.

Moreover, according to your kind advice, we refer to other literature for the method section as follows.

References “[58]”, ”[59]”, “[60]”, “[61]”, “[62]”, “[63]”, “[64]” and “[69]” were added in Page 21 Line 540, Page 21 Line 542, Page 21 Line 543, Page 21 Line 550, Page 21 Line 555, Page 21 Line 561, Page 22 Line 586, and Page 23 Line 657, respectively. Other references were listed in order.

According to the requirements and formats of our journal, we added the cited literature in References section:

  1. Pattemore,A. RNA extraction from cereal vegetative tissue. Methods Mol.Biol. 2004, 1099, 17–21.
  2. Wei, Q.;Jiao, C.; Guo, L.; Ding, Y.; Cao, J.; Feng, J.; Dong, X.; Mao, L.; Sun, H.; Yu, F.; Yang, G.; Shi, P.; Ren, G.; Fei, Z. Exploring key cellular processes and candidate genes regulating the primary thickening growth of Moso underground shoots. New Phytol. 2017, 241, 81–96.
  3. Davies, J.;Denyer, T.; Hadfield, J. Bioanalyzer chips can be used interchangeably for many analyses of DNA or RNA. BioTechniques 2016, 60, 197–199.
  4. Anderson, A.J.;Culver, H.R.; Prieto, T.R.; Martinez, P.J.; Sinha, J.; Bryant, S.J.; Bowman, C.N. Messenger RNA enrichment using synthetic oligo(T) click nucleic acids. Chem. Commun. 2020, 56, 13987–13990.
  5. Degenkolbe, T.;Hannah, M.A.; Freund, S.; Hincha, D.K.; Heyer, A.G.; Köhl, K.I. A quality-controlled microarray method for gene expression profiling. Anal. Biochem. 2005, 346, 217–224.
  6. Annaluru, N.;Muller, H.; Ramalingam, S.; Kandavelou, K.; London, V.; Richardson, S.M.; Dymond, J.S.; Cooper, E.M.; Bader, J.S.; Boeke, J.D.; Chandrasegaran, S. Assembling DNA fragments by USER fusion. Methods Mol. Biol. 2012, 852, 77–95.
  7. Kanehisa,M.; Araki, M.; Goto, S.; Hattori, M.; Hirakawa, M.; Itoh, M.; Katayama, T.; Kawashima, S.; Okuda, S.; Tokimatsu, T.; Yamanishi, Y. KEGG for linking genomes to life and the environment. Nucleic Acids Res. 2008, 36, 480–484.
  8. Qi, N.;Hou, X.; Wang, C.; Li, C.; Huang, D.; Li, Y.; Wang, N.; Liao, W. Methane-rich water induces bulblet formation of scale cuttings in Lilium davidii unicolor by regulating the signal transduction of phytohormones and their levels. Physiol. Plant. 2021, 172, 1919–1930.

Point 8: The figures’ resolution needs to be improved and clear, currently, all the figures are not in good resolution.

Response 8: Thank you for your kind suggestion. We increased each figure resolution to 600dpi in accordance with your advice in the revised manuscript.

Point 9: Since it is an English paper, the sentences should be smooth and reasonable, and it is best to find an English-speaking expert to check.

Response 9: Thank you for contributing so much to improving our manuscript. Our manuscript has undergone English language editing by MDPI according to your comment. The text has been checked for correct use of grammar and common technical terms, and edited to a level suitable for reporting research in a scholarly journal. The certificate of English language editing is in the following:

Finally, we highly appreciate the editors and reviewers who took the time to offer their careful criticism and advice. We would like to thank the detailed valuable comments of the referees on our manuscript. The suggestions help us improve it to a better scientific level.

This manuscript is a resubmission of an earlier submission. The following is a list of the peer review reports and author responses from that submission.

Round 1

Reviewer 1 Report

This paper provides a useful transcriptomic data to contribute to understand the mechanism of adventitious root formation in cucumber, and has the potential to be acceptable for IJMS. The paper can be accepted after the following revisions are carried out appropriately.

L500: in 100 ml of RNase-free water? 100 μl?

L106-113, L343-350, Figure 1C: I recommend that you keep the order of terms consistent.

Figure 1C: The gene ratio by the X-axis and the size of the dot are redundant. The data could be summarized as a table.

Conclusion: You cannot really say that those DEGs are positively or negatively associated with the ETH-induced adventitious rooting. Those genes are ethylene-responsive genes, and then, the “candidate” genes associated with the ETH-induced adventitious rooting.

Author Response

Point 1: This paper provides a useful transcriptomic data to contribute to understand the mechanism of adventitious root formation in cucumber, and has the potential to be acceptable for IJMS. The paper can be accepted after the following revisions are carried out appropriately.

Response 1: Thank you very much for your comments. We are pleased to have been given the opportunity to revise our manuscript. We have incorporated your specific comments in the preparation of the revised version of our manuscript. We have adequately addressed your concerns and we hope that the manuscript is now acceptable for publication in “International Journal of Molecular Sciences”. If you have any questions, please feel free to contact us. We look forward to receiving your decisions.

Point 2: L500: in 100 ml of RNase-free water? 100 μl?

Response 2: We appreciate your insightful comments. Thank you most sincerely for pointing out this error to the improvement of this manuscript. We revised this sentence in the following.

“Total RNA was dissolved in 100 μL of RNase-free water and was quantified using a NanoDrop spectrophotometer (Thermo Scientific).” (Please see Page 20 Lines 515-516)

Point 3: L106-113, L343-350, Figure 1C: I recommend that you keep the order of terms consistent.

Response 3: Thank you for your good suggestion. According to your comment, we revised the order of top 20 KEGG pathways in the revised manuscript in the following. Moreover, in discussion section, we identified common KEGG entries in proteome and transcriptome data, and the order was ranked by transcriptome data. Thus, To give readers a better understanding of our results, we added an explanation there.

“The top 20 KEGG pathways accordance with the most abundant DEGs were displayed in Table 1. Among them, “linoleic acid metabolism (6 DEGs)”, “starch and sucrose metabolism (17 DEGs)”, “sesquiterpenoid and triterpenoid biosynthesis (5 DEGs)”, “fatty acid degradation (8 DEGs)”, “isoquinoline alkaloid biosynthesis (4 DEGs)”, “phenylalanine metabolism (7 DEGs)”, “plant hormone signal transduction (26 DEGs)”, “fatty acid metabolism (9 DEGs)”, “pentose and glucuronate interconversions (10 DEGs)” and “cutin, suberine and wax biosynthesis (5 DEGs)” were top 10 enriched terms in KEGG pathways (Table 1 and Supplementary Table S4).” (Please see Page 3 Lines 107-114)

“In top 20 KEGG pathway terms, “starch and sucrose metabolism”, “phenylalanine metabolism”, “plant hormone signal transduction”, “fatty acid metabolism”, “pentose and glucuronate interconversions”, “cutin, suberine and wax biosynthesis”, “photosynthesis - antenna proteins”, “vitamin B6 metabolism”, “purine metabolism”, and “alanine, aspartate and glutamate metabolism” were both enriched in transcriptome and proteome (the order was ranked by transcriptome data),”  (Please see Page 17 Lines 353-359)

Point 4: Figure 1C: The gene ratio by the X-axis and the size of the dot are redundant. The data could be summarized as a table.

Response 4: We appreciate the reviewer’s insightful advice. KEGG pathway analysis was summarized as Table 1 as follows. (Please see Page 5 Line 123)

Table 1 Enriched KEGG pathways of the DEGs in ETH treatment.

KEGGID

Pathway

pvalue

Total DEGs

Up-regulated DEGs

Down-regulated DEGs

csv00591

Linoleic acid metabolism

0.006637889

6

4

2

csv00500

Starch and sucrose metabolism

0.007112864

17

2

15

csv00909

Sesquiterpenoid and triterpenoid biosynthesis

0.007603753

5

2

3

csv00071

Fatty acid degradation

0.017687328

8

6

2

csv00950

Isoquinoline alkaloid biosynthesis

0.018821384

4

4

0

csv00360

Phenylalanine metabolism

0.021066898

7

5

2

csv04075

Plant hormone signal transduction

0.026635092

26

13

13

csv01212

Fatty acid metabolism

0.032091296

9

7

2

csv00040

Pentose and glucuronate interconversions

0.037498837

10

2

8

csv00073

Cutin, suberine and wax biosynthesis

0.039280085

5

0

5

Point 5: Conclusion: You cannot really say that those DEGs are positively or negatively associated with the ETH-induced adventitious rooting. Those genes are ethylene-responsive genes, and then, the “candidate” genes associated with the ETH-induced adventitious rooting.

Response 5: Thank you very much for you to review and give valuable advice. We revised this part in the following.

“In conclusion, the candidate genes of sucrose and starch metabolism (CsHK2), glycolysis-related gene (CsPK2), phenylalanine metabolism (CsPAL), flavonoid biosynthesis (CsF3M), auxin signaling transduction (CsLAX5, CsGH3.17 and CsSUAR50) and ETH signaling transduction (CsERS) were ETH-down regulated genes, whereas the candidate genes of starch metabolism (CsBAM1 and CsBAM3), phenylalanine metabolism (CsPAO), flavonoid biosynthesis (CsHCT), ABA signaling transduction (CsPYL1, CsPYL5 and CsPYL8) and BR signaling transduction (CsBAK1 and CsXTH3) were up-regulated in ETH-induced adventitious rooting. These candidate genes were associated with the ETH-induced adventitious root development. This study will be useful to understand the molecular mechanisms of ETH-inducing adventitious rooting.”

(Please see Page 23 Lines 652-661)

Reviewer 2 Report

In this manuscript, the authors report in detail the key genes involved in ethylene-induced adventitious root development in cucumber seedlings. The text is generally well written and can be recommended for publication in IJMS. Some minor comments are:

L.62-64. “Under water-logging, both auxin and ETH inhibitor significantly prevented the adventitious rooting in cucumber [17].” The reference is wrong. Bai et al. [17] described a study with an apple rootstock.

L.480. It is desirable to indicate the variety of cucumber with which the experiments were carried out.

The quality of the figures is low, the text on them is hard to see.

Are there similar studies with other species from the Cucurbitaceae family?

Author Response

Point 1: In this manuscript, the authors report in detail the key genes involved in ethylene-induced adventitious root development in cucumber seedlings. The text is generally well written and can be recommended for publication in IJMS.

Response 1: Thanks very much for your attention and helps on our paper. Authors would like to thank you for the decision that our manuscript can be recommended for publication in your journal after revision. We have incorporated your specific comments in the preparation of the revised version of our manuscript.

Point 2: L.62-64. “Under water-logging, both auxin and ETH inhibitor significantly prevented the adventitious rooting in cucumber [17].” The reference is wrong. Bai et al. [17] described a study with an apple rootstock.

Response 2: Thank you for the valuable comment. As you suggested, We re-read the refrenece again and revised this sentence as follows.

“Both auxin and ETH inhibitor significantly prevented the adventitious rooting in apple [17].” (Please see Page 2 Lines 63-64)

Point 3: L.480. It is desirable to indicate the variety of cucumber with which the experiments were carried out.

Response 3: We really appreciate for contributing such a good idea to the improvement of this manuscript. We added the variety of cucumber with which the experiments were carried out in the revised manuscript.

“The cucumber (C. sativus L. “Xinchunsihao”) seeds, soaking in distilled water for 5-6 h, with the same size and full grains were chosen and put on a tray with wet filter paper. “ (Please see Page 20 Lines 494-496)

Point 4: The quality of the figures is low, the text on them is hard to see.

Response 4: Thank you for your kind suggestion. We redrew all figures in accordance with your advice in the revised manuscript.

Point 5: Are there similar studies with other species from the Cucurbitaceae family?

Response 5: Thank you very much. We have searched the references about ETH-induced adventitious root development in other species from the Cucurbitaceae family. However, there is no relevant reference at present.

Reviewer 3 Report

 In the manuscript "Key genes during ethylene-induced adventitious root development in cucumber (Cucumis sativus L.)’  Deng and collaborators investigated the genes that might be involved in the development of adventitious roots in cucumber.

Overall, the manuscript should be revised in terms of writing. There are several inaccuracies, specially in Introduction section. The abstract is clear and concise and presents an overview of the results and points out the main conclusions and importance of the study. The Materials and Methods section should be also revised. For example:

·       Line 607: Poorly described, is missing the following information:

o   Amount of RNA used in reverse transcription

o   Amount of cDNA used in qPCR reaction

o   Primers concentration used in qPCR reaction

o   Software used to design the primers

o   Whether no-template controls (NTCs) were used to assess contaminations and primer dimers formation

o   Whether PCR efficiencies of amplicons were calculated by performing a standard curve

o   Whether melting curve analysis was done to ensure amplification of the specific amplicon

o   Refence of the 2-delta delat Ct

o   Number of technical and biological replicates used

o   The software used to acquire Ct/Cq values

·       Line 609: change to: …Gene expression data was normalized using Csactin as …

·       Line 610: change to: … 2-delta delta Ct was used to calculate normalized gene expression levels.’

·       Line 612: change to: ‘The thermocycling conditions were: 95 °C for 30 s and 40 cycles of 95 °C for 5 s, 60 °C for 30 s.’

·       Line 612: Table S2 should has NCBI gene accession numbers, product size, primers efficiency, correlation coefficient

·       Line 613: change to: ‘ RT-qPCR reactions were performed on a ABI stepone plus (California, USA). Remove the rest of the sentence.

·       Line 616: change to: ‘statistical analysis’ instead

Additionally, replace qRT-PCR to RT-qPCR instead throughout the text.

Author Response

Point 1: In the manuscript "Key genes during ethylene-induced adventitious root development in cucumber (Cucumis sativus L.)’ Deng and collaborators investigated the genes that might be involved in the development of adventitious roots in cucumber.

Overall, the manuscript should be revised in terms of writing. There are several inaccuracies, specially in Introduction section. The abstract is clear and concise and presents an overview of the results and points out the main conclusions and importance of the study. The Materials and Methods section should be also revised.

Response 1: We highly appreciate you who took the time to offer your careful criticism and advice. We would like to thank the detailed valuable comments of you on our manuscript. The suggestions help us improve it to a better scientific level. We have adequately revised the errors in the revised manuscript as below. We look forward to receiving your decisions.

Point 2: Line 607: Poorly described, is missing the following information:

o   Amount of RNA used in reverse transcription

o   Amount of cDNA used in qPCR reaction

o   Primers concentration used in qPCR reaction

o   Software used to design the primers

o   Whether no-template controls (NTCs) were used to assess contaminations and primer dimers formation

o   Whether PCR efficiencies of amplicons were calculated by performing a standard curve

o   Whether melting curve analysis was done to ensure amplification of the specific amplicon

o   Refence of the 2-delta delat Ct

o   Number of technical and biological replicates used

o   The software used to acquire Ct/Cq values

Response 2: We appreciate your insightful comments. Thank you most sincerely for putting forward such a great idea to the improve our manuscript. In accordance with your suggestion, we revised this part as below:

“In accordance with the manufacturer’s instructions, RNA was reverse transcribed with 5 × Evo M-MLVRT Master Mix (AG, Hunan, China). Extracted RNA (400 ng) was reverse transcribed to cDNA with AMV Reverse Transcriptase (Life Science Advanced Technology, St Petersburg, FL, USA). SYBR Green Pro Taq HS Premix (AG, Hunan, China) was used to amplify cDNA. The amount of cDNA used in each RT-qPCR reaction was: 1 μL for target gene, 1 μL for reference genes, and 0.6 μL of 1:100 diluted cDNA for 16S rRNA. Gene expression data was normalized using Csactin as internal control. [62]. The primers was designed in (https://pga.mgh.harvard.edu/primerbank/). 20 μL of PCR reaction was prepared with 1X SYBR Green, 1X ROX dye (Roche, Basel, Switzerland), 1 μM forward and reverse primer. No-template controls (NTCs) were not used to assess contaminations and primer dimers formation. RT-qPCR efficiencies of amplicon were calculated by performing a standard curve. Moreover, melting curve analysis was done to ensure amplification of the specific amplicon. 2-ΔΔct was used to calculate normalized gene expression levels.. The reference of 2-ΔΔct was 0. The primers are shown in Supplementary Table S2. The reaction control conditions were: 95 °C for 30 s and 40 cycles of 95 °C for 5 s, 60 °C for 30 s. RT-qPCR reactions were performed on a ABI stepone plus (California, USA). Each treatment had 3 technical and biological replicates replicates. Ct/Cq values were acquired by LightCycler® 96 System.” (Please see Pages 22-23 Lines 627-644)

Point 3: Line 609: change to: …Gene expression data was normalized using Csactin as …

Response 3: Thank you for your insight suggestion. The sentence was changed as below:

“Gene expression data was normalized using Csactin as internal control. [62].” (Please see Page 22 Lines 633-634)

Point 4: Line 610: change to: … 2-delta delta Ct was used to calculate normalized gene expression levels.’

Response 4: We appreciate your good advice. The sentence was revised in the following:

“2-ΔΔct was used to calculate normalized gene expression levels.” (Please see Page 23 Lines 639-640)

Point 5: Line 612: change to: ‘The thermocycling conditions were: 95 °C for 30 s and 40 cycles of 95 °C for 5 s, 60 °C for 30 s.’

Response 5: Thank you for the valuable comment. According to your suggestion, this sentence was changed in the revised manuscript in the following:

“The primers are shown in Supplementary Table S2. The reaction control conditions were: 95 °C for 30 s and 40 cycles of 95 °C for 5 s, 60 °C for 30 s.” (Please see Page 23 Lines 641-642)

Point 6: Line 612: Table S2 should has NCBI gene accession numbers, product size, primers efficiency, correlation coefficient

Response 6: Thank you for your valuable comment. According to your advice, we added NCBI gene accession numbers, product size, primers efficiency and correlation coefficient in Table S2 in the following.

Supplementary Table S2 Sequences of primers used for RT-qPCR analysis

Gene symbol

NCBI gene accession numbers

Gene ID

Product size

Primer Sequence (5'-3' )

Primers efficiency

Correlation coefficient (R2)

CsGBE1

ACHR03000062

101215687

107

F:GACAGTGAAGGGTTGGCAGGTTG

R:TCATCATCTTCGGGTTTGCTCGTATC

89.64%

0.9853

Cseg9

ACHR03000028

101222401

81

F:GTTACTAGCGACGGCGAAGAAGG

R:ACAGCAGAGGAAAGCGAATCACTATAC

90.38%

0.9942

Cseg6

ACHR03000050

101203271

111

F:CAGTGGTGGACATAGTGGCTACAATC

R:GAAGACGATGCTGGAGATGGAAGTG

89.27%

0.9837

Csβ-G11-like

ACHR03000050

101214838

116

F:CTTTCGGTTTGTTCCTTTGACTGACTC

R:ACTTCGGATACTCACCATACACCAATG

94.69%

0.9918

CsHK2

ACHR03000083

101216058

113

F:TACCTTGGTGAAATTGCTCGTAGAGTG

R:TCTGGTGTGCTCAGGATGAATTGC

93.76%

0.9969

CsSS5-like

ACHR03000028

101211461

91

F:TTTACGACGAAGAATGGGCAAATGATG

R:GATGGAAGGCTTAACCGAGGAGTTG

90.47%

0.9893

CsPFP

ACHR03000058

101215219

83

F:TTCTCCCGCCAATCCTCCTTCTC

R:GATGGATACGATGGGCTTGGACTTG

93.66%

0.9921

CsPK2

ACHR03000062

101206620

81

F:CTCGTGTGGTAGACAGCATGACTAAC

R:CATCAAGTACAGCATTGGCAACATCAG

94.28%

0.9915

CsACS2

ACHR03000062

101209539

83

F:GATGAAAGAGGTGAGGAGCCAAGATAC

R:ATAACATCGCCAACGCCATAGCC

90.79%

0.9863

CsIDH

ACHR03000028

101202770

105

F:TCCTCCACCTCCTCCTCTACTCTC

R:AGAAGAAGAAGAAGAAGCGTAGCATCG

96.88%

0.9960

CsLACS6

ACHR03000083

101207461

82

F:TTAATTGCGAGTGTTGCTGGATGTTC

R:GTGCCAAAGGGAGGTACGATATGTAG

95.77%

0.9868

CsCYP450 86A1

ACHR03000058

101214516

96

F:CGCTGGACCGAGGACTTGTTTG

R:ACTGGAGATAGCCGATACCGAAGG

99.06%

0.9947

CsLAX5

ACHR03000006

101205750

144

F:TCTACATCATTCCTGCTCTTGCTCAC

R:CCAACACCCAAACCACTACAAATGC

94.65%

0.9774

CsGH3.17

ACHR03000050

101206247

84

F:TGAAAGAATTGCCAATGGTGAACCTTC

R:CCAGAAGTTCCTGAGCTTGTGAGAAG

97.14%

0.9927

CsSAUR50

ACHR03000083

101217643

81

F:TGTCTACGTTGGTCAACACCGAAC

R:TGGAGCAAGATTTGGAAAGGAGGATG

96.57%

0.9886

CsPYR1

ACHR03000083

101204882

96

F:GACGAGTTGAAGGACTTAGTAGCAGAG

R:TGAACACGCTGAGCAAGTAAGGAG

95.80%

0.9914

CsCS-ERS

ACHR03000006

101205786

131

F:AAGTTGCTTGTGCTATTGTATCGTGTG

R:GCCCATCTCCCTGTCAAGTTGTTC

101.42%

0.9861

CsERFC3

ACHR03000014

101206564

129

F:CGGAGATACGAGACTCAACCAGAAATG

R:AATTAAGAACGGCGGCGGAACC

97.75%

0.9975

CsBAK1

ACHR03000006

101216386

132

F:GGGTCGTTTGGCTGATGGTTCTC

R:AGACGGAGTAGATTACGGTGGACAG

93.30%

0.9879

CsPAL

ACHR0300006

101218856

148

F:AGCATCATCCTGGACAGATTGAAGC

R:TGCGGTGAAGTTCTAAGAGCGTAAC

98.49%

0.9934

CsPAO

ACHR03000028

101213059

136

F:CCTTCTAGTCTCTCGTTTGTGGTTTCC

R:AGTGAGTGGATCTAGTGGATGGTGAG

103.43%

0.9948

CsAt4g34880

ACHR03000006

101204792

136

F:GCTGACAGAGAACGAGAGGCTAATAAG

R:TAGACCCAAGCAATGCGAATGATCC

89.24%

0.9715

Csnad5

HQ860792

11123919

148

F:CGCACAGATAGGATCGCATACTTGG

R:AGCCGTAGGTGGGTATTCAAATAAAGG

99.03%

0.9951

CsF3M

ACHR03000058

101207160

103

F:ATCGTGGAGGAGCATCGGAATTTG

R:TCTCATCTTCACCGTCATCCTTCATTG

105.32%

0.9649

CsHCT

ACHR03000014

101207100

135

F:ACCCTGAGTTTCTGAGACAATTCCAAG

R:CGTCATAAAGAGGAAGTCGGCAGAG

87.63%

0.9802

Point 7: Line 613: change to: ‘ RT-qPCR reactions were performed on a ABI stepone plus (California, USA). Remove the rest of the sentence.

Response 7: We appreciate your kind advice. In accordance, the sentence was revised in the following:

“RT-qPCR reactions were performed on a ABI stepone plus (California, USA).” (Please see Page 23 Lines 642-643)

Point 8: Line 616: change to: ‘statistical analysis’ instead

Response 8: Thank you very much for pointing out this error in our manuscript. The title of 4.8 was changed to “Statistical analysis”. (Please see Page 23 Line 645)

Point 9: Additionally, replace qRT-PCR to RT-qPCR instead throughout the text.

Response 9: We appreciate your insightful comments. Thank you most sincerely for contributing so much to the improvement of this manuscript. qRT-PCR was replace throughout the revised manuscript in accordance with your suggestion as follows:

“qRT-PCR” was changed to “RT-qPCR” in Page 6 Line153, Page 10 Line 229, Page 20 Line 513 and Page 22 Line 626, respectively.

Round 2

Reviewer 3 Report

 In the revised manuscript "Key genes during ethylene-induced adventitious root development in cucumber (Cucumis sativus L.)’  Deng and collaborators did not improve the RT-qPCR procedure in the material and methods section, still presenting several incorrections.

For example:

·       Line 630: ‘….The amount of cDNA used in each RT-qPCR reaction was: 1 μL for target gene, 1 μL for reference genes, and 0.6 μL of 1:100 diluted cDNA for 16S rRNA.’ … the sentence makes no sense;

·        Line 634: ‘… for the PCR reaction preparation there is no need to place here the ROX passive reference, but instead the authors should place the cDNA amount used, which is missing;

·       Line 636: the authors say that ‘No-template controls (NTCs) were NOT used’, however NTCs should be always used in order to assess contaminations and primer dimers formation.

·       Line 640: the authors say that ‘the reference of 2-ΔΔct was 0’. This makes no sense. I meant the bibliographic reference. The authors should become aware of the reference ‘Livak KJ, Schmittgen TD. Analysis of relative gene expression data using real-time quantitative PCR and the 2(-Delta Delta C(T)) Method. Methods. 2001 Dec;25(4):402-8. doi: 10.1006/meth.2001.1262. which should be placed in the bibliographic references and when citing the formula 2-ΔΔct.

·       Line 642: what the authors mean with ‘reaction control conditions’?... replace to ‘cycling conditions’ instead

·       Line 647: just ‘Ct values’ since previously Ct was used

I deeply advise the authors to consult a scientist with experience in this technology to help the authors revising this part of the manuscript.

Author Response

Response to Reviewer 3 Comments

Point 1: In the revised manuscript ‘Key genes during ethylene-induced adventitious root development in cucumber (Cucumis sativus L.)’ Deng and collaborators did not improve the RT-qPCR procedure in the material and methods section, still presenting several incorrections.

Response 1: Thanks very much for your attention and helps on our paper. We would like to thank you for the decision that our manuscript can be recommended for publication in your journal after revision. We have incorporated your specific comments about RT-qPCR procedure in the preparation of the revised version of our paper.

Point 2: Line 630: ‘….The amount of cDNA used in each RT-qPCR reaction was: 1 μL for target gene, 1 μL for reference genes, and 0.6 μL of 1:100 diluted cDNA for 16S rRNA.’ … the sentence makes no sense;

Response 2: Thank you for the valuable comment. As you suggested, this sentence was deleted in the revised manuscript.

Point 3: Line 634: ‘… for the PCR reaction preparation there is no need to place here the ROX passive reference, but instead the authors should place the cDNA amount used, which is missing;

Response 3: We really appreciate for contributing such a good idea to the improvement of this manuscript. We added the missing information in the revised version.

“The amount of cDNA in each treatment was 2-3 μL. The rate of 260/280 was around 1.7. The concentration the rate of 260/280 was quantified using NanoDrop 2000 Ultra Micro Volume Spectrophotometer (Thermo Scientific, USA)” was added in the revision. (Please see Page 22 Lines 630-633)

Additionally, the sentence “20 μL of PCR reaction was prepared with 1X SYBR Green, 1X ROX dye (Roche, Basel, Switzerland), 1 μM forward and reverse primer” was removed according to your suggestion.

Point 4: Line 636: the authors say that ‘No-template controls (NTCs) were NOT used’, however NTCs should be always used in order to assess contaminations and primer dimers formation.

Response 4: Thank you for your kind suggestion. This is such a valuable idea to improve our manuscript. As you suggested, we redid the experiment with no-template controls (NTCs). The revised parts were in the following:

“In order to assess contaminations and primer dimers formation and verify the reliability of the data, no-template controls (NTCs) were used [63]” was added in the revision. (Please see Page 23 Lines 637-639)”

“63.Stals, A.; Werbrouck, H.; Baert, L., Botteldoorn, N.; Herman, L.; Uyttendaele, M.; Van Coillie, E. Laboratory efforts to eliminate contamination problems in the real-time RT-PCR detection of noroviruses. J Microbiol Methods 2009, 77, 72-6. (Please see Page 25 Lines 808-809)”

Furthermore, we revised the primers efficiency and the content of correlation coefficient (R2) in Supplementary Table S2.

Point 5: Line 640: the authors say that ‘the reference of 2-ΔΔct was 0’. This makes no sense. I meant the bibliographic reference. The authors should become aware of the reference ‘Livak KJ, Schmittgen TD. Analysis of relative gene expression data using real-time quantitative PCR and the 2(-Delta Delta C(T)) Method. Methods. 2001 Dec;25(4):402-8. doi: 10.1006/meth.2001.1262. which should be placed in the bibliographic references and when citing the formula 2-ΔΔct.

Response 5: Thank you very much. We re-read the reference that you provided, and added this reference in the revised text in the following:

“2-ΔΔct was used to calculate normalized gene expression levels [64].” (Please see Page 23 Lines 641-642)”

“64.Livak, K.J.; Schmittgen, T.D. Analysis of relative gene expression data using real-time quantitative PCR and the 2(-Delta Delta C(T)) Method. Methods 2001, 25, 402-8. (Please see Page 25 Lines 810-811)”

Point 6: Line 642: what the authors mean with ‘reaction control conditions’?... replace to ‘cycling conditions’ instead;

Response 6: Thank you very much. We revised this error in the revised manuscript in the following:

“The cycling conditions were: 95 °C for 30 s and 40 cycles of 95 °C for 5 s, 60 °C for 30 s. (Please see Page 23 Lines 642-643)”

Point 7: Line 647: just ‘Ct values’ since previously Ct was used.

Response 7: I agreed with you. Thus, the sentence was revised as follows:

“Ct values were acquired by LightCycler® 96 System. (Please see Page 23 Lines 645-646)”

Point 8: I deeply advise the authors to consult a scientist with experience in this technology to help the authors revising this part of the manuscript.

Response 8: We are appreciate for your valuable advice. Therefore, we consulted with professors and staff who were familiar with RT-qPCR procedure. They gave us some advice to improve this part. Some method such as “using no-template controls (NTCs)” could extremely improve the primers efficiency, and we will use it in our further experiment. We added the PCR reaction system in the revised manuscript in the following:

“Then, 7.2 μL ddH2O, 0.4 μL forward primer, 0.4 μL reverse primer, 2 μL cDNA and 10 μL Taq enzyme was added to a sterilized PCR tube.” (Please see Pages 22-23 Lines 635-637)”

Moreover, we know about the procedure of RT-qPCR now. If we still have some missing information, please let us know.

Round 3

Reviewer 3 Report

In the second revision of the manuscript "Key genes during ethylene-induced adventitious root development in cucumber (Cucumis sativus L.)’  Deng and collaborators, unfortunately, did not improve the description of RT-qPCR methodology, keeping describing the methodology in an incomprehensible way. Once the manuscript is mostly about the ‘key genes’, which should be accurately validated by RT-qPCR, the fact that this technique contains many flaws make me to consider that the manuscript is not suitable for publication.

For example:

·       L629: the authors say: The amount of cDNA in each treatment was 2-3 μL…. And then in L636 the authors say they used 2ul cDNA… I suggested to place the cDNA amount in nanograms taking into account the amount of RNA (400 ng) reverse transcribed and the final volume of cDNA solution

·       L630: the authors say: The rate of 260/280 in each treat,ent was around 1.7. …. This makes no sense.

·       L630: the authors say: The concentration the rate of 260/280 was quantified using ……. This makes no sense.

·       L645: the authors say: RT-qPCR reactions were performed on a ABI stepone plus …. But then in L649 they say: Ct values were acquired by LightCycler® 96 System…. Did the authors make the reactions in one equipment and acquired the Ct values with another equipment?.... this makes no sense.